# Multi-Objective Reinforcement Learning for Forward-Backward Markov Decision Processes

## Abstract

This work introduces the notion of Forward-Backward Markov Decision Process (FB-MDP) for multi-task control problems. In this context, we devise a novel approach called Forward-Backward Multi-Objective Reinforcement Learning (FB-MORL). Specifically, we analytically characterize its convergence towards a Pareto-optimal solution and also empirically evaluate its effectiveness. For the latter, we consider a use case in wireless caching and perform several experiments to characterize performance in that context. Finally, an ablation study demonstrates that FB-MDP is instrumental to optimize rewards for systems with forward-backward dynamics. The outcomes of this work pave the way for further understanding of multi-objective RL algorithms for FB-MDPs.

## 1 Introduction

Reinforcement Learning (RL) is very important in the field of artificial intelligence, as it enables intelligent agents to learn from experience and adapt to complex, dynamic environments (Mnih et al., 2013; Lillicrap et al., 2016; Schulman et al., 2017). Moreover, recent breakthroughs in deep reinforcement learning (DRL) – namely, RL leveraging deep neural networks – led to intelligent policies that exceed human-level performance in wide variety of domains (Mnih et al., 2015; Jaderberg et al., 2018; Rigoli et al., 2021).

Existing RL algorithms mainly address sequential decision-making problems modeled as a forward Markov decision process or controlled forward dynamics. However, there are several sequential tasks whose environment cannot be exclusively captured by this type of dynamics, as they also encompass states that evolve according to a backward dynamics. The pricing and hedging of a European option can be framed as a backward dynamics problem, showing the application of these dynamics in determining optimal investment and risk management strategies (Chessari et al., 2023). Such dynamics provide a trajectory that starts from the current state and follows a specific action to transition to a preceding state. Moreover, the backward dynamics admits a final known state, in contrast to the forward dynamics with is based on an initial state. Dynamics in a form of (controlled) backward stochastic differential equation (BSDE) have wide applications in stochastic control (Zhang, 2017) and (differential) game theory (Hamadène & Lepeltier, 2000; Grun, 2012; Zhang, 2022), as well as in mathematical finance (El Karoui et al., 1997; Bouchard et al., 2018; Hientzsch, 2019). In the context of stochastic optimal control, Pontryagin's maximum principle and Feynman-Kac representation of Partial Differential Equations (PDEs) are a few examples of using BSDE (Yong & Zhou, 1999), whereas pricing problems and hedging theory are sample applications in the domain of mathematical finance (Ma & Yong, 1999). There are also problems that can be modeled based on a controlled forward-backward dynamics, or alternatively, as a Forward-Backward Markov decision process (FB-MDP). In the latter case, both the forward and backward process coexist and conflict with each other through the action space at the same time. Here, we present a motivating example of backward MDPs in the context of computation offloading (Zabihi et al., 2023). Imagine a mobile computing unit that processes offloaded tasks using its computational resources while buffering them as needed. The average time for successful computation is influenced by factors such as the probability of buffer overflow and the processing time of buffered tasks. In the case of overflow, tasks are re-offloaded at this unit. Consequently, it results in a backward MDP linked to action parameters including buffer and computation capacity (see Appendix E for more details).

Controlled forward-backward dynamics have been extensively studied in the context of stochastic differential equations. However, these dynamics have not been adequately addressed in the domain of Markov processes and problems based on reinforcement learning. The primary challenge arises from the interplay and conflict between the forward and backward processes within the action space. More specifically, an optimal policy needs to jointly consider both forward and backward processes. However, this contradicts the causal nature of the forward dynamics and the backward dynamics evolving in a time-reversed trajectory.

Motivated by such a challenge, we introduce the notion of FB-MDPs for multi-task problems entailing conflicting forward and backward rewards. Accordingly, we propose a novel multi-objective RL algorithm called Forward-Backward Multi-Objective Actor-Critic (FB-MOAC), which is built upon the Advantage Actor-Critic (A2C) framework (Grondman et al., 2012). Moreover, we conduct a rigorous convergence analysis of FB-MOAC by specifying the conditions to achieve convergence-on-mean. To the best of our knowledge, this is the first study presenting an RL algorithm for a class of FB-MDPs, coupled with a comprehensive analysis of its convergence and performance in a selected use case. In summary, the contributions of this work are the following.

(1) We formalize a forward-backward multi-task Markov decision process, recognize a class of action-coupled FB-MDP for which we derive the backward Bellman's and Pareto-optimality equations, and develop FB-MOAC, a multi-objective RL algorithm for such settings.

(2) We provide a novel theoretical framework based on which we analytically characterize FB-MOAC with a convergence-on-mean guarantee towards Pareto-optimal solutions.

(3) We assess the performance of FB-MOAC through an experimental evaluation in the context of wireless caching, inclusive of an ablation study.

**Notation**: $\|\mathbf{A}\|$ is the induced matrix norm of $\mathbf{A}$. We use $\mathbf{I}_n$, $\mathbf{1}_n$, $\mathbf{0}$ and $\mathbf{e}_m$ to denote the identity matrix of size $n \times n$, a $n$-dimensional vector with all elements equal to one, a vector with all elements equal to zero, and a vector with all elements being zero except the $m$-th element that is one, respectively. We use $|S|$ to shows the cardinality of the set $S$, and $[\,\cdot\,]$ to indicate the components of row vectors.

## 2 PRELIMINARIES

We introduce next the notion of Pareto optimality and forward-backward Markov decision processes.

### 2.1 PARETO OPTIMALITY

We briefly introduce the notion of Pareto optimality for a multi-objective optimization (MOO) problem. Accordingly, consider the following unconstrained problem:

$$Q_1: \quad \min_{\boldsymbol{x} \in \mathcal{X}} \; [f_1(\boldsymbol{x}), \ldots, f_r(\boldsymbol{x})],$$

where $f_j : \mathbb{R}^N \to \mathbb{R}$, $\mathcal{X}$ is the feasible set, and $r$ is the number of objectives. Then, $\boldsymbol{x}^* \in \mathcal{X}$ is called a Pareto optimal solution of $Q_1$, if there is no other solution $\boldsymbol{y} \in \mathcal{X}$ so as to dominate $\boldsymbol{x}^*$, i.e., $f_i(\boldsymbol{y}) \leq f_i(\boldsymbol{x}^*)$ for all $i \in \{1, \ldots, r\}$ and there is one $j$ such that $f_j(\boldsymbol{y}) < f_j(\boldsymbol{x}^*)$.

If there exists a vector $\boldsymbol{\alpha} \in [0,1]^r$ with $\sum_{j=1}^r \alpha_j = 1$ so that $\sum_{j=1}^r \alpha_j \nabla f_j(\hat{\boldsymbol{x}}) = 0$, then $\hat{\boldsymbol{x}}$ is a Pareto optimal solution and $[f_1(\hat{\boldsymbol{x}}), \ldots, f_r(\hat{\boldsymbol{x}})]$ is a Pareto front for MOO $Q_1$.

The following Lemma (Schäffler et al., 2002; Ma et al., 2020) provides guidance on jointly reducing all objectives of MOO $Q_1$.

**Lemma 2.1.** *Assume a multi-valued multivariate function $\boldsymbol{f} = (f_1, \ldots, f_r)$, $f_j : \mathbb{R}^n \to \mathbb{R}$ for $j \in \{1, \ldots, r\}$. Define $\boldsymbol{q}(\cdot) = \sum_{j=1}^r \alpha_j^* \nabla f_j(\cdot)$, then $-\boldsymbol{q}(\cdot)$ is a descent direction for all functions $\{f_j(\cdot)\}_1^r$, where $\{\alpha_j^*\}$ are the solution of the following optimization problem:*

$$Q_2: \quad \min_{\{\alpha_j\}_1^r} \; \left\| \sum_{j=1}^r \alpha_j \nabla f_j(\cdot) \right\|^2, \quad \text{s.t.} \quad \sum_{j=1}^r \alpha_j = 1, \quad \alpha_j \geq 0 \quad j \in \{1, \ldots, r\}.$$

Accordingly, the optimal solution of problem $Q_2$ can be obtained using the following Corollary.

**Corollary 2.1.** *The solution of $Q_2$, for $\nabla\boldsymbol{f}(\cdot)^\top\nabla\boldsymbol{f}(\cdot)$ being invertible and with all $\alpha_j \geq 0$ is:*

$$\boldsymbol{\alpha}^* = \left(\mathbf{1}_r^\top\left(\nabla\boldsymbol{f}(\cdot)^\top\nabla\boldsymbol{f}(\cdot)\right)^{-1}\mathbf{1}_r\right)^{-1}\left(\nabla\boldsymbol{f}(\cdot)^\top\nabla\boldsymbol{f}(\cdot)\right)^{-1}\mathbf{1}_r, \tag{1}$$

*where $\nabla\boldsymbol{f}(\cdot)$ is an $n \times r$ matrix with $\nabla\boldsymbol{f}(\cdot) = [\nabla f_1, \ldots, \nabla f_r](\cdot)$. For the case $\alpha_j < 0$ for $j \in \mathcal{S}_0 \in \{1,\ldots,r\}$, we set $\nabla\boldsymbol{f}(\cdot) = [\nabla f_k(\cdot)]_{k \in \{1,\ldots,r\}\setminus\mathcal{S}_0}$.*

## 2.2 Forward-Backward Markov Decision Process

An FB-MDP is expressed by a tuple $(\mathcal{S}, \mathcal{Y}, \mathcal{A}, P_f(\cdot), P_b(\cdot), \boldsymbol{r}(\cdot))$, where: $\mathcal{S}$ and $\mathcal{Y}$ are the forward and backward state spaces, respectively; $\mathcal{A}$ is the action space; $P_f(\cdot): \mathcal{S} \times \mathcal{A} \times \mathcal{S} \rightarrow [0,1]$ is the forward transition probability, which describes the forward dynamics; $P_b(\cdot): \mathcal{Y} \times \mathcal{A} \times \mathcal{Y} \rightarrow [0,1]$ is the backward transition probability, which expresses the backward dynamics; and $\boldsymbol{r}(\cdot): \mathcal{S} \times \mathcal{Y} \times \mathcal{A} \rightarrow \mathbb{R}^m$ is the immediate multivariate reward function, where $m \in \mathbb{N}$ denotes the dimension of the reward function. Being at the forward state $\mathbf{s}_t \in \mathcal{S}$ and performing the action $\mathbf{a}_t \in \mathcal{A}$, the forward transition probability probabilistically determines the next forward state of the system $\mathbf{s}_{t+1} \sim P_f(\cdot|\mathbf{s}_t, \mathbf{a}_t)$. Moreover, in an anti-causal way, being at the backward state $\mathbf{y}_t \in \mathcal{Y}$ and performing the action $\mathbf{a}_t \in \mathcal{A}$, the previous backward state of the system follows: $\mathbf{y}_{t-1} \sim P_b(\cdot|\mathbf{y}_t, \mathbf{a}_t)$. Notice that the initial forward state $\mathbf{s}_1$ and final backward state $\mathbf{y}_T$ are known. Furthermore, the forward and backward transition probabilities are coupled, as they depend on a common action space.

In this paper, we introduce a class of FB-MDPs with forward rewards $\boldsymbol{r}^f(\cdot): \mathcal{S} \times \mathcal{A} \rightarrow \mathbb{R}^{|S_f|}$ and backward rewards $\boldsymbol{r}^b(\cdot): \mathcal{Y} \times \mathcal{A} \rightarrow \mathbb{R}^{|S_b|}$, where $S_f$ and $S_b$ are the sets of indexes of forward and backward rewards, respectively. Moreover, the backward and forward rewards are coupled merely within the action space. We term this class of processes as *action-coupled FB-MDPs*. The aim of this FB-MDP problem is thus to optimize the following multi-objective discounted cumulative reward from the Pareto-optimality perspective:

$$\max_{\{\mathbf{a}_t \in \mathcal{A}\}_{t \in [1,T]}} \mathbb{E}\left\{\sum_{t=1}^{T}\gamma^{t-1}\left[\boldsymbol{r}^f(\mathbf{s}_t, \mathbf{a}_t), \ \boldsymbol{r}^b(\mathbf{y}_{T-t+1}, \mathbf{a}_{T-t+1})\right]\right\}, \tag{2}$$

where $T \in \mathbb{N}$ is the optimization finite horizon, $\gamma \in (0,1]$ the discount factor, and the expectation is with respect to the different realizations of the forward-backward trajectory $\boldsymbol{\tau}: \mathbf{s}_1 \rightarrow \mathbf{a}_1 \rightarrow \mathbf{s}_2 \rightarrow \ldots \rightarrow \mathbf{a}_T, \ \mathbf{y}_T \rightarrow \mathbf{a}_T \rightarrow \mathbf{y}_{T-1} \rightarrow \ldots \rightarrow \mathbf{a}_1 \rightarrow \mathbf{y}_1$.

## 3 Forward-Backward Multi-Objective RL Algorithm

Note that the backward states $\{\mathbf{y}_{T-t+1}\}_{t \in [1,T]}$ cannot be revealed before actions $\{\mathbf{a}_t\}_{t \in [1,T]}$ are designed, and actions should be optimized by considering both the forward and backward rewards. To tackle this and solve problem (2), we devise a forward-backward step-wise mechanism explained in Table (1). According to this mechanism and due to the Markov property, the probability of the

Table 1: Forward-Backward Step-wise Mechanism

| | |
|---|---|
| **Step 1** | **Forward Evaluation** |
| | Consider a $\boldsymbol{\theta}$-parametric stochastic policy distribution $\pi_{\boldsymbol{\theta}}(\cdot|\mathbf{s}_t)$. |
| | Generate action $\mathbf{a}_t \sim \pi_{\boldsymbol{\theta}}(\cdot|\mathbf{s}_t)$ and evaluate $\mathbf{s}_{t+1} \sim P_f(\cdot|\mathbf{s}_t, \mathbf{a}_t)$, for $t \in [1, T-1]$. |
| **Step 2** | **Backward Evaluation** |
| | Evaluate $\mathbf{y}_{T-t} \sim P_b(\cdot|\mathbf{y}_{T-t+1}, \mathbf{a}_{T-t+1})$ for $t \in [1, T-1]$ based on the generated actions of the previous step. |
| **Step 3** | **Forward-Backward Optimization** |
| | Optimize the policy distribution $\pi_{\boldsymbol{\theta}}(\cdot|\cdot)$ based on the evaluated forward and backward rewards using a multi-objective optimization. |

trajectory $\boldsymbol{\tau}$:

$$
\begin{array}{c}
\overbrace{\mathbf{s}_1 \to \mathbf{a}_1 \to \mathbf{s}_2 \to \ldots \mathbf{a}_t \ldots \to \quad \mathbf{s}_T \quad \to \mathbf{a}_T}^{\text{forward direction}} \\
\updownarrow \qquad\qquad \updownarrow \qquad\qquad\qquad \updownarrow \\
\underbrace{\mathbf{a}_1 \leftarrow \mathbf{y}_1 \leftarrow \ldots \mathbf{a}_t \ldots \leftarrow \mathbf{y}_{T-1} \leftarrow \mathbf{a}_T \leftarrow \mathbf{y}_T}_{\text{backward direction}}
\end{array}
$$

is determined by:

$$
\mathrm{P}_{\boldsymbol{\theta}}(\boldsymbol{\tau}) := \mathbb{P}(\mathbf{s}_1, \mathbf{a}_1, \ldots, \mathbf{s}_t, \mathbf{a}_t, \mathbf{y}_T, \ldots, \mathbf{y}_1)
$$

$$
= \underbrace{\mathbb{P}(\mathbf{s}_1) \prod_{t=1}^{T-1} P_f(\mathbf{s}_{t+1}|\mathbf{s}_t, \mathbf{a}_t)}_{\mathrm{P}^f(\boldsymbol{\tau})} \times \underbrace{\mathbb{P}(\mathbf{y}_T) \prod_{t=1}^{T-1} \pi_{\boldsymbol{\theta}}(\mathbf{a}_t|\mathbf{s}_t) P_b(\mathbf{y}_{T-t}|\mathbf{y}_{T-t+1}, \mathbf{a}_{T-t+1})}_{\mathrm{P}'_{\boldsymbol{\theta}}(\boldsymbol{\tau})} = \mathrm{P}^f(\boldsymbol{\tau}) \, \mathrm{P}'_{\boldsymbol{\theta}}(\boldsymbol{\tau}). \quad (3)
$$

Therefore, problem (2) can be reformulated as following *policy distribution optimization*:

$$
\mathrm{O}_2: \quad \max_{\pi_{\boldsymbol{\theta}}(\cdot|\mathbf{s}_t)} \mathbb{E}_{\mathrm{P}_{\boldsymbol{\theta}}(\boldsymbol{\tau})} \left\{ \sum_{t=1}^{T} \gamma^{t-1} \left[ \boldsymbol{r}^f(\mathbf{s}_t, \mathbf{a}_t), \ \boldsymbol{r}^b(\mathbf{y}_{T-t+1}, \mathbf{a}_{T-t+1}) \right] \,\Big|\, \boldsymbol{\theta} \right\}
$$

$$
\text{s.t.} \quad \left\{ \mathbf{s}_{t+1} \sim P_f(\cdot|\mathbf{s}_t, \mathbf{a}_t), \ \mathbf{y}_{t-1} \sim P_b(\cdot|\mathbf{y}_t, \mathbf{a}_t), \ \mathbf{a}_t \sim \pi_{\boldsymbol{\theta}}(\cdot|\mathbf{s}_t) \right\}. \quad (4)
$$

The multivariate objective of $\mathrm{O}_2$ can thus be expressed as:

$$
\mathbf{J}(\boldsymbol{\theta}) := \Big[ \underbrace{\mathbb{E}_{\mathrm{P}_{\boldsymbol{\theta}}(\boldsymbol{\tau})} \sum_{k=1}^{T} \gamma^{k-1} \boldsymbol{r}^f(\mathbf{s}_k, \mathbf{a}_k)}_{\mathbf{J}_f(\boldsymbol{\theta})}, \ \underbrace{\mathbb{E}_{\mathrm{P}_{\boldsymbol{\theta}}(\boldsymbol{\tau})} \sum_{k=1}^{T} \gamma^{k-1} \boldsymbol{r}^b(\mathbf{y}_{T-k+1}, \mathbf{a}_{T-k+1})}_{\mathbf{J}_b(\boldsymbol{\theta})} \Big].
$$

To maximize $\mathbf{J}(\boldsymbol{\theta})$, we need its gradient with respect to $\boldsymbol{\theta}$, i.e., $\nabla_{\boldsymbol{\theta}} \mathbf{J}(\boldsymbol{\theta}) = \frac{\partial \mathbf{J}(\boldsymbol{\theta})}{\partial \mathrm{P}_{\boldsymbol{\theta}}(\boldsymbol{\tau})} \frac{\partial \mathrm{P}_{\boldsymbol{\theta}}(\boldsymbol{\tau})}{\partial \boldsymbol{\theta}}$. We then compute the gradient of $\mathbf{J}(\boldsymbol{\theta})$ component-wise. For the forward cumulative rewards $\mathbf{J}_f(\boldsymbol{\theta})$, we have (Grondman et al., 2012):

$$
\nabla_{\boldsymbol{\theta}} \mathbf{J}_f(\boldsymbol{\theta}) = \mathbb{E} \left\{ \sum_{k=1}^{T} \nabla_{\boldsymbol{\theta}} \log \pi_{\boldsymbol{\theta}}(\mathbf{a}_k|\mathbf{s}_k) \mathbf{A}^f(\mathbf{s}_k, \mathbf{a}_k) \,\Big|\, \boldsymbol{\theta} \right\}, \quad (5)
$$

where $\mathbf{A}^f(\cdot, \cdot) : \mathcal{S} \times \mathcal{A} \to \mathbb{R}^{|S_f|}$, $\mathbf{A}^f(\mathbf{s}_k, \mathbf{a}_k) := \boldsymbol{r}^f(\mathbf{s}_k, \mathbf{a}_k) + \gamma V^f(\mathbf{s}_{k+1}) - V^f(\mathbf{s}_k)$ is the forward advantage multivariate function and $V^f(\cdot) : \mathcal{S} \to \mathbb{R}^{|S_f|}$, $V^f(\mathbf{s}_k) := \mathbb{E} \left\{ \sum_{k'=k}^{T} \gamma^{k'-k} \boldsymbol{r}^f(\mathbf{s}_{k'}, \mathbf{a}_{k'})|\mathbf{s}_k \right\}$ is the forward state-value multivariate function. We then have the following lemma which characterizes the optimal backward trajectories.

**Lemma 3.1.** *For the backward cumulative reward $\boldsymbol{J}_b(\boldsymbol{\theta})$, we can get:*

$$
\nabla_{\boldsymbol{\theta}} \boldsymbol{J}_b(\boldsymbol{\theta}) = \mathbb{E}_{P_{\boldsymbol{\theta}}(\tau)} \left\{ \sum_{k=0}^{T-1} \nabla_{\boldsymbol{\theta}} \log \pi_{\boldsymbol{\theta}}(\boldsymbol{a}_{T-k}|\boldsymbol{s}_{T-k}) \boldsymbol{A}^b(\boldsymbol{y}_{T-k}, \boldsymbol{a}_{T-k}) \,\Big|\, \boldsymbol{\theta} \right\} \quad (6)
$$

*where $A^b(\cdot, \cdot) : \mathcal{Y} \times \mathcal{A} \to \mathbb{R}^{|S_b|}$ is the backward advantage multivariate function:*

$$
A^b(\boldsymbol{y}_{T-k}, \boldsymbol{a}_{T-k}) := \boldsymbol{r}^b(\boldsymbol{y}_{T-k}, \boldsymbol{a}_{T-k}) + \gamma V^b(\boldsymbol{y}_{T-k-1}) - V^b(\boldsymbol{y}_{T-k}),
$$

*and $V^b(\cdot) : \mathcal{Y} \to \mathbb{R}^{|S_b|}$ is the backward state-value multivariate function:*

$$
V^b(\boldsymbol{y}_{T-k}) := \mathbb{E} \left\{ \sum_{k'=k}^{T-1} \gamma^{k'-k} \boldsymbol{r}^b(\boldsymbol{y}_{T-k'}, \boldsymbol{a}_{T-k'})|\boldsymbol{y}_{T-k} \right\},
$$

*which adheres to the following backward Bellman's equation:*

$$
V^b(\boldsymbol{y}_{T-k}) = \mathop{\mathbb{E}}_{\substack{\boldsymbol{a}_{T-k} \sim \pi_{\boldsymbol{\theta}}(\cdot|\boldsymbol{s}_{T-k}) \\ \boldsymbol{y}_{T-k-1} \sim P_b(\cdot|\boldsymbol{y}_{T-k}, \boldsymbol{a}_{T-k})}} \left\{ \boldsymbol{r}^b(\boldsymbol{y}_{T-k}, \boldsymbol{a}_{T-k}) + \gamma V^b(\boldsymbol{y}_{T-k-1}) \,\big|\, \boldsymbol{\theta} \right\}. \quad (7)
$$

*Furthermore, a Bellman Pareto-optimality equation can be derived as:*

$$
\left[ V^{f^*}(\boldsymbol{s}), V^{b^*}(\boldsymbol{y}) \right] = \max_{\boldsymbol{a}} \left[ \mathop{\mathbb{E}}_{s^+ \sim P_f(\cdot|s,a)} \left\{ \boldsymbol{r}^f(\boldsymbol{s}, \boldsymbol{a}) + \gamma V^{f^*}(\boldsymbol{s}^+) \right\}, \mathop{\mathbb{E}}_{y^- \sim P_b(\cdot|y,a)} \left\{ \boldsymbol{r}^b(\boldsymbol{y}, \boldsymbol{a}) + \gamma V^{b^*}(y^-) \right\} \right], \quad (8)
$$

*for $(\boldsymbol{s}, \boldsymbol{y}, \boldsymbol{a}) \in \mathcal{S} \times \mathcal{Y} \times \mathcal{A}$, where $\left( V^{f^*}(\boldsymbol{s}), V^{b^*}(\boldsymbol{y}) \right)$ is a Pareto front, $\boldsymbol{s}^+ \in \mathcal{S}$ is the forward state following $\boldsymbol{s}$, and $\boldsymbol{y}^- \in \mathcal{Y}$ is the backward state preceding $\boldsymbol{y}$.*

*Proof.* Please refer to Appendix B. □

**Remarks.** The formulations of Lemma 3.1 differs from their counterparts in forward dynamics. Specifically, Eq. (7) exhibits a forward dynamics with a dependency on the backward transition probability as well as on the policy distribution that itself relies on the forward state rather than the backward state. Moreover, the Bellman's Pareto-optimality equation Eq. (8) describes an optimum solution for the action-coupled FB-MDPs and necessitates the usage of a multi-objective optimization. It is noteworthy that this Lemma is specifically applicable to the action-coupled FB-MDPs.

According to Eqs. (5) and (6), we need to evaluate the policy distribution $\pi_{\boldsymbol{\theta}}(\cdot|\cdot)$ and the state-value functions $V^f(\cdot)$ and $V^b(\cdot)$ to compute the gradient of $\mathbf{J}(\boldsymbol{\theta})$. For the policy distribution $\pi_{\boldsymbol{\theta}}(\cdot|\cdot)$, we establish an *actor agent* represented by a $\boldsymbol{\theta}$-parameterized NN. For the forward state-value function $V^f(\cdot)$, we set a *forward-critic agent* represented by a $\boldsymbol{\phi}$-parametric NN, and denoted as $V^f_{\boldsymbol{\phi}}(\cdot)$. Moreover, we use a *backward-critic agent* with a $\boldsymbol{\psi}$-parametric NN for backward state-value function indicated by $V^b_{\boldsymbol{\psi}}(\cdot)$. We must now align the evaluation and update procedures for actor and critic agents with the proposed forward-backward mechanism in Table (1). In this regard, $\pi_{\boldsymbol{\theta}}(\cdot|\cdot)$ and $V^f_{\boldsymbol{\phi}}(\cdot)$ are evaluated during the forward-evaluation step of the proposed mechanism, $V^b_{\boldsymbol{\psi}}(\cdot)$ is evaluated during the backward-evaluation step, and their values are leveraged to compute $\nabla_{\boldsymbol{\theta}}\mathbf{J}(\boldsymbol{\theta})$ and update $\pi_{\boldsymbol{\theta}}(\cdot|\cdot)$ during the forward-backward optimization step.

Since the update mechanism of actor policy $\pi_{\boldsymbol{\theta}}(\cdot|\cdot)$ depends on the forward and backward state-value functions, i.e., $V^f_{\boldsymbol{\phi}}(\cdot)$ and $V^b_{\boldsymbol{\psi}}(\cdot)$, we need to set some losses to tune these state-value functions. In the line with Bellman's equation $V^f(\mathbf{s}_k) = \mathbb{E}_{\mathbf{s}_{k+1},\mathbf{a}_k|\mathbf{s}_k}\{r^f(\mathbf{s}_k,\mathbf{a}_k) + \gamma V^f(\mathbf{s}_{k+1})\}$ and Temporal Difference (TD)-learning (Grondman et al., 2012), the following *forward-critic losses* are considered to update parameter $\boldsymbol{\phi}$:

$$A^f_{\boldsymbol{\phi},i}(\mathbf{s}_k,\mathbf{a}_k)^2 = \left(V^f_{\boldsymbol{\phi},i}(\mathbf{s}_k) - r^f_i(\mathbf{s}_k,\mathbf{a}_k) - \gamma V^f_{\boldsymbol{\phi},i}(\mathbf{s}_{k+1})\right)^2, \qquad i \in S_f, \tag{9}$$

where $\{A^f_{\boldsymbol{\phi},i}(\cdot) : \mathcal{S} \times \mathcal{A} \to \mathbb{R}\}_{i \in S_f}$ are parametric representations for the forward advantage functions. Conversely, we set the following *backward-critic losses* to update parameter $\boldsymbol{\psi}$ based on the derived Bellman's equation Eq. (7):

$$A^b_{\boldsymbol{\psi},i}(\mathbf{y}_{T-k},\mathbf{a}_{T-k})^2 = \left(V^b_{\boldsymbol{\psi},i}(\mathbf{y}_{T-k}) - r^b_i(\mathbf{y}_{T-k},\mathbf{a}_{T-k}) - \gamma V^b_{\boldsymbol{\psi},i}(\mathbf{y}_{T-k-1})\right)^2, \quad i \in S_b, \tag{10}$$

where $\{A^b_{\boldsymbol{\psi},i}(\cdot) : \mathcal{Y} \times \mathcal{A} \to \mathbb{R}\}_{i \in S_b}$ are parametric representations for the backward advantage functions.

Recalling Eqs. (5), (6), (9) and (10), which indicate how the policy distribution and forward/backward state-value functions should be updated, a multi-objective loss needs to be addressed for each of them. One straightforward approach to cope with this issue, namely the scalarization technique, is to obtain a single-objective loss by considering a preference function (or scales) for different losses. However, the Pareto solutions cannot be necessarily obtained via this method (Kirlik & Sayın, 2014). As a consequence, a trial-and-error approach might be needed to tune the scalarization settings, which makes this approach sensitive to the selected setup. Instead, we use a scalar-independent multi-objective optimization method (Schäffler et al., 2002) to devise a forward-backward RL algorithm. Accordingly, we leverage Lemma 2.1 to formulate forward/backward critic agents and a multi-objective actor agent shared between forward and backward critics. It is detailed in the next sections.

## 3.1 FORWARD/BACKWARD CRITIC AGENTS

In the light of Lemma 2.1, we formulate a multi-objective forward-critic loss $K^f(\boldsymbol{\phi})$ as well as a multi-objective backward-critic loss $K^b(\boldsymbol{\phi})$. Considering Eqs. (10) and (9), we thus set:

$$K^f(\boldsymbol{\phi}) = \sum_{j \in S_f} \sum_{k=1}^{T} \beta^*_{f,j} A^f_{\boldsymbol{\phi},j}(\mathbf{s}_k,\mathbf{a}_k)^2, \qquad K^b(\boldsymbol{\psi}) = \sum_{j \in S_b} \sum_{k=0}^{T-1} \beta^*_{b,j} A^b_{\boldsymbol{\psi},j}(\mathbf{y}_{T-k},\mathbf{a}_{T-k})^2, \tag{11}$$

where the vectors $\boldsymbol{\beta}^*_f$ and $\boldsymbol{\beta}^*_b$ are optimized by

$$\boldsymbol{\beta}_f^* = \underset{\substack{\beta_j \geq 0 \\ \sum_{j \in S_f} \beta_j = 1}}{\operatorname{argmin}} \left\| \sum_{j \in S_f} \beta_j \nabla_{\boldsymbol{\phi}} \sum_{k=1}^{T} A_{\boldsymbol{\phi},j}^f(\mathbf{s}_k, \mathbf{a}_k)^2 \right\|^2, \qquad \boldsymbol{\beta}_b^* = \underset{\substack{\beta_j \geq 0 \\ \sum_{j \in S_b} \beta_j = 1}}{\operatorname{argmin}} \left\| \sum_{j \in S_b} \beta_j \nabla_{\boldsymbol{\psi}} \sum_{k=0}^{T-1} A_{\boldsymbol{\psi},j}^b(\mathbf{y}_{T-k}, \mathbf{a}_{T-k})^2 \right\|^2.$$

(12)

Then, the forward-critic and backward-critic agents are updated by the TD-learning with the following stochastic gradient descent (SGD) rules (Grondman et al., 2012):

$$\boldsymbol{\phi} \leftarrow \boldsymbol{\phi} - \mu_f \nabla_{\boldsymbol{\phi}} K^f(\boldsymbol{\phi}), \qquad \boldsymbol{\psi} \leftarrow \boldsymbol{\psi} - \mu_b \nabla_{\boldsymbol{\psi}} K^b(\boldsymbol{\psi}),$$

(13)

where $\mu_f$ and $\mu_b$ are the learning rates of the forward and backward agents, respectively.

## 3.2 ACTOR AGENT

To derive the actor loss, we follow the same strategy with a minor modification. According to Eqs. (5) and (6), the following forward and backward gradients are first set:

$$\nabla_{\boldsymbol{\theta}} \hat{J}_j^f(\boldsymbol{\theta}, \boldsymbol{\phi}) = -\sum_{k=1}^{T} \nabla_{\boldsymbol{\theta}} \log \pi_{\boldsymbol{\theta}}(\mathbf{a}_k|\mathbf{s}_k) A_{\boldsymbol{\phi},j}^f(\mathbf{s}_k, \mathbf{a}_k), \qquad j \in S_f,$$

$$\nabla_{\boldsymbol{\theta}} \hat{J}_j^b(\boldsymbol{\theta}, \boldsymbol{\psi}) = -\sum_{k=1}^{T} \nabla_{\boldsymbol{\theta}} \log \pi_{\boldsymbol{\theta}}(\mathbf{a}_k|\mathbf{s}_k) A_{\boldsymbol{\psi},j}^b(\mathbf{y}_k, \mathbf{a}_k), \qquad j \in S_b.$$

(14)

Then, the multi-objective actor agent is updated by the following SGD:

$$\boldsymbol{\theta} \leftarrow \boldsymbol{\theta} - \mu \Big( \sum_{j \in S_f} \beta_{\text{act,j}} \nabla_{\boldsymbol{\theta}} \hat{J}_j^f(\boldsymbol{\theta}, \boldsymbol{\phi}) + \sum_{j \in S_b} \beta_{\text{act,j}} \nabla_{\boldsymbol{\theta}} \hat{J}_j^b(\boldsymbol{\theta}, \boldsymbol{\psi}) \Big),$$

(15)

where $\mu$ is the learning rate of actor agent, and $\{\beta_{\text{act,j}}\}$ are obtained by:

$$\boldsymbol{\beta}_{\text{act,j}} = \underset{\substack{\beta_j \geq 0 \\ \sum_{j \in S_f \cup S_b} \beta_j = 1}}{\operatorname{argmin}} \left\| \sum_{j \in S_f} \beta_j \nabla_{\boldsymbol{\theta}} \bar{J}_j^f(\boldsymbol{\theta}) + \sum_{j \in S_b} \beta_j \nabla_{\boldsymbol{\theta}} \bar{J}_j^b(\boldsymbol{\theta}) \right\|^2,$$

(16)

with

$$\nabla_{\boldsymbol{\theta}} \bar{J}_j^f(\boldsymbol{\theta}) := -\mathbb{E}_{\boldsymbol{\phi}} \mathbb{E} \left\{ \sum_{k=1}^{T} \nabla_{\boldsymbol{\theta}} \log \pi_{\boldsymbol{\theta}}(\mathbf{a}_k|\mathbf{s}_k) A_{\boldsymbol{\phi},j}^f(\mathbf{s}_k, \mathbf{a}_k) \,\Big|\, \boldsymbol{\theta} \right\}, \qquad j \in S_f,$$

$$\nabla_{\boldsymbol{\theta}} \bar{J}_j^b(\boldsymbol{\theta}) := -\mathbb{E}_{\boldsymbol{\psi}} \mathbb{E} \left\{ \sum_{k=1}^{T} \nabla_{\boldsymbol{\theta}} \log \pi_{\boldsymbol{\theta}}(\mathbf{a}_k|\mathbf{s}_k) A_{\boldsymbol{\psi},j}^b(\mathbf{y}_k, \mathbf{a}_k) \,\Big|\, \boldsymbol{\theta} \right\}, \qquad j \in S_b.$$

(17)

Note that, as opposed to the critic losses, we theoretically leverage the expected gradient $\nabla_{\boldsymbol{\theta}} \bar{J}_j^f(\boldsymbol{\theta})$ and $\nabla_{\boldsymbol{\theta}} \bar{J}_j^b(\boldsymbol{\theta})$ to optimize $\boldsymbol{\beta}_{\text{act}}$ in Eq. (16) [see Eq. (12)]. However, to estimate each of them, we practically employ *Monte Carlo Sampling* (MCS) accompanied with a *exponential moving average* applied to $\nabla_{\boldsymbol{\theta}} \hat{J}_j^f(\boldsymbol{\theta}, \boldsymbol{\phi})$ and $\nabla_{\boldsymbol{\theta}} \hat{J}_j^b(\boldsymbol{\theta}, \boldsymbol{\psi})$, respectively. More specifically, we first implement $N_{\text{mcs}}$ distinct backward and forward critic networks with learned parameters $\{\boldsymbol{\psi}_l\}_{l=1}^{N_{\text{mcs}}}$ and $\{\boldsymbol{\phi}_l\}_{l=1}^{N_{\text{mcs}}}$, respectively, and we then use the approximations $\mathbb{E}_{\boldsymbol{\phi}}\{F(\boldsymbol{\phi})\} \approx \frac{1}{N_{\text{mcs}}} \sum_{l=1}^{N_{\text{mcs}}} F(\boldsymbol{\phi}_l)$ and $\mathbb{E}_{\boldsymbol{\psi}}\{G(\boldsymbol{\psi})\} \approx \frac{1}{N_{\text{mcs}}} \sum_{l=1}^{N_{\text{mcs}}} G(\boldsymbol{\psi}_l)$ with $F(\cdot)$ and $G(\cdot)$ representing the relevant objectives of interest. In addition, we consider different episodes of the algorithm and take an exponential average over them with a smoothing factor $\gamma_{\text{mov}}$. Notice that this moving average approximates the inner expectation in Eq. (17). We name this approach **episodic MCS-average** standing for MCS of the actor stochastic objectives and the moving average.

Figure 3 and Algorithm 1 in Appendix D overviews the proposed Forward-Backward Multi-Objective Actor-Critic (FB-MOAC) algorithm and shows its pseudo-code, respectively.

## 4 EVALUATION

### 4.1 ANALYTICAL RESULTS

We prove that the FB-MOAC algorithm is guaranteed to find (i) a Pareto-optimal solution with a convergence rate of $\mathcal{O}(1/K)$ for the strongly-convex and Lipschitz smooth case and (i) a locally

Pareto-optimal solution with a convergence rate of $\mathcal{O}(1/\sqrt{K})$ for the Lipschitz-smooth case, where $K$ is the number of policy updates. The related proofs are available in Appendix C. These results are aligned with those for single-optimization algorithms for forward-MDPs (Fu et al., 2021).

## 4.2 EXPERIMENTAL RESULTS

We conduct experiments for the following objectives: evaluate the FB-MOAC algorithm on two real-world FB-MDP problems with conflicting forward and backward reward; characterize the impact of the backward optimization procedure on learning.

### 4.2.1 EVALUATION ON FORWARD-BACKWARD MULTI-TASK PROBLEMS

**Hybrid Delivery Scheme:** Here, we exploit a forward-backward multi-task problem in the context of wireless caching to empirically evaluate the developed FB-MOAC RL algorithm. Notice that this problem is in contrast to numerous conventional RL problems, as it incorporates a coupled forward-backward dynamics, a model often absent in the standard RL problems. For this, we adapt the system model presented in (Amidzadeh et al., 2022). Note that, we use this problem not only because its environment relies on a FB-MDP, but also because it represents a real-world sequential decision-making task.

The related environment is a cellular network assisted by two distinct types of serving nodes, namely base-stations (BSs) and helper-nodes (HNs). These nodes are spatially distributed across the network with intensities $\lambda_{\mathrm{bs}}$ and $\lambda_{\mathrm{hn}}$, respectively. It also includes mobile users that request content from the cellular network, spatially distributed with an intensity $\lambda_{\mathrm{ue}}$. Each user requests a file out of $N$ distinct contents with different popularity. The cellular system operates over time-slots with index $t \in \{1, \ldots, T\}$, where $T$ is the total duration within which the network operation is considered. At each time-slot, the requests of users from the network can be modeled based on time-varying dynamics, and the purpose of the network is to satisfy as many users as possible through both the BSs and HNs. For this, the network applies multicast and unicast transmission schemes on the HNs and BSs, respectively. The HNs proactively cache most popular files and cooperatively broadcast the cached files across the network at the beginning of each time-slot using some controlling parameters, denoted by the vector $\boldsymbol{p}^{\mathrm{MC}}(t)$. However, a multicast outage may occur with probability $\{O_n^{\mathrm{MC}}(\boldsymbol{p}^{\mathrm{MC}}(t), t)\}_{n=1}^N$, resulting in certain users not being satisfied in receiving their requested content. These unsatisfied users then request the content through the BSs to be served by the unicast transmission. The unicast unit then exploits the controlling parameters $\boldsymbol{p}^{\mathrm{UC}}(t)$ to satisfy requesting users. Similarly, an unicast outage may occur with probability $O^{\mathrm{UC}}(\boldsymbol{p}^{\mathrm{UC}}(t), t)$. Users not satisfied by this hybrid transmission scheme (i.e., the combination of multicast and unicast) will have their requested contents deferred to the next time-slot. Therefore, at each time-slot there is a distribution of users accounting for the repeated requests and a distribution describing the new preferences. This leads to a time-varying model for the request probability of content $n$ denoted by $q_n(t)$:

$$q_n(t) = b_n(t) \sum_{m=1}^N (1 - O_m^{\mathrm{tot}}(t-1)) q_m(t-1) + q_n(t-1) O_n^{\mathrm{tot}}(t-1), \quad n \in \{1, \ldots, N\}, \qquad (18)$$

where $O_n^{\mathrm{tot}}(t) = O_n^{\mathrm{MC}}(\boldsymbol{p}^{\mathrm{UC}}(t), t) \, O^{\mathrm{UC}}(\boldsymbol{p}^{\mathrm{UC}}(t), t)$ is the total outage probability, and $b_n(t)$, as a priori information, stands for a network-wide content popularity (Sadeghi et al., 2018) of file $n$. Note that Eq. (18) represents a **forward dynamics**, with the forward state vector $\mathbf{s}(t) = \mathbf{q}(t)$ and the action vector $\mathbf{a}(t) = [\boldsymbol{p}^{\mathrm{MC}}(t), \boldsymbol{p}^{\mathrm{UC}}(t)]$ affecting $O_n^{\mathrm{tot}}(\cdot)$.

A file request is repeated across several time-slots until successful reception, resulting in an expected latency $L_n(t)$ for successful reception of file $n$. For this quantity, a time-varying dynamics can be derived by the law of total expectation as follows:

$$L_n(t) = O_n^{\mathrm{tot}}(t) \Big( d(t) + L_n(t+1) \Big) + \big( 1 - O_n^{\mathrm{MC}}(\boldsymbol{p}^{\mathrm{MC}}(t), t) \big) \frac{1}{2} d(t), \quad L_n(T) = 0, \quad n \in \{1, \ldots, N\}, \quad (19)$$

where $d(t)$ is the duration of time-slot $t$ in seconds. Note that Eq. (19) represents a **backward dynamics**, with the backward state vector $\mathbf{y}(t) = \mathbf{L}(t)$ and the action vector $\mathbf{a}(t)$. Thus, Eqs. (18) and (19) together can be modeled by a FB-MDP, and should be jointly considered to derive an optimal content delivery scheme. Further, it portrays a coupled process as the forward and backward dynamics are coupled with each other through the multicast and unicast parameters $\mathbf{a}(t)$ (which is the common action vector).

| Hyper-parameter | Value |
|---|---|
| Learning rates of | |
| Actor and Critics: | $2 \times 10^{-4}$ |
| Number of MCS $N_{\mathrm{MCS}}$: | 4 |
| Smoothing factor $\gamma_{\mathrm{mov}}$: | 0.95 |
| Number of neurons | |
| in hidden layers | |
| of Actor and Critics: | 100 |

(a)                                   (b)                                   (c)

Figure 1: FB-MOAC Hyper-parameters. Performance results of FB-MOAC on (b) the hybrid scheme and (c) the multicast scheme.

Three widely-used *network performance metrics* (Li et al., 2018) (terminologically interpreted as reward functions) are considered to design an optimum policy: quality-of-service (QoS) $r_{\mathrm{QoS}}(\cdot)$, total bandwidth consumption $r_{\mathrm{BW}}(\cdot)$ and overall expected latency $r_{\mathrm{Lat}}(\cdot)$. For the QoS, the total intensity of unsatisfied UEs is:

$$r_{\mathrm{QoS}}(t) = \sum_{n=1}^{N} q_n(t) O_n^{\mathrm{tot}}(t), \tag{20}$$

whereas the total bandwidth consumption is:

$$r_{\mathrm{BW}}(t) = W^{\mathrm{MC}}\big(\boldsymbol{p}^{\mathrm{MC}}(t), t\big) + W^{\mathrm{UC}}\big(\boldsymbol{p}^{\mathrm{UC}}(t), t\big), \tag{21}$$

where $W^{\mathrm{MC}}(\boldsymbol{p}^{\mathrm{MC}}(t), t)$ and $W^{\mathrm{UC}}(\boldsymbol{p}^{\mathrm{UC}}(t), t)$ represent the total bandwidth consumption for the multicast and unicast transmission, respectively. Finally, the overall expected latency is:

$$r_{\mathrm{Lat}}(t) = \sum_{n=1}^{N} q_n(t) L_n(t), \tag{22}$$

with $L_n(t)$ obtained through Eq. (19). Clearly $r_{\mathrm{QoS}}(t)$ and $r_{\mathrm{BW}}(t)$ relate to the forward state, and constitute a forward bivariate reward function $\boldsymbol{r}^f(t) = [r_{\mathrm{QoS}}(t), r_{\mathrm{BW}}(t)]$. Since $r_{\mathrm{Lat}}(t)$ relates to the backward state, it instead constitutes a backward reward function $r^b(t) = r_{\mathrm{Lat}}(t)$.

Consequently, a forward-backward multi-task problem, called *hybrid experiment*, with three competing objectives is formulated, and we thus apply Algorithm 1 to find a dynamic solution.

**Multicast Delivery Scheme:** Here, we further introduce another FB-MDP problem. In this scenario, the network exclusively applies the multicast unit without using unicast transmissions. As such, the probability of unicast outage becomes one: $O^{\mathrm{UC}}(\boldsymbol{p}^{\mathrm{UC}}(t), t) = 1$ and no bandwidth is consumed by this unit: $W^{\mathrm{UC}}(\boldsymbol{p}^{\mathrm{UC}}(t), t) = 0$. Accordingly, the forward and backward dynamics presented in Eqs. (18) and (19) will change. This gives rise to a new problem, called the multicast experiment, with distinct system models, action parameters, reward functions, and MDPs.

### 4.2.2 SETUP AND HYPER-PARAMETERS

We consider the following settings for the considered system environment (Amidzadeh et al., 2022). The number of files is $N = 100$, the spatial intensities of BSs, HNs and users are $\lambda_{\mathrm{bs}} = 10$, $\lambda_{\mathrm{hn}} = 100$ and $\lambda_{\mathrm{ue}} = 10^5$, respectively, in points/km$^2$. The desired rate of transmission is set to 1 Mbps, this quantity affects the multicast and unicast outage probability. The total number of time-slots is $T = 256$ and the discount factor $\gamma = 0.96$.

As for the *FB-MOAC* algorithm, three separate sets of NNs represent the multi-objective actor, forward-critic and backward-critic agents. Moreover, we use $N_{\mathrm{MCS}}$ many NNs for the forward-critic agent as well as for the backward-critic one. The multi-objective forward-critic network outputs two values representing the reward-specific state-value functions $V_{\phi,j}^f(\cdot)$ which are related to the QoS $r_{\mathrm{QoS}}(\cdot)$ and total bandwidth consumption $r_{\mathrm{BW}}(\cdot)$. On the other hand, the backward-critic network outputs one value representing the state-value functions $V_{\psi}^b(\cdot)$ which is related to the overall latency $r_{\mathrm{Lat}}(\cdot)$. For each NN, one hidden layer is considered and the rectified linear unit (ReLU) activation function is used for the neuron connectivities. The other hyper-parameters are in Table 1a.

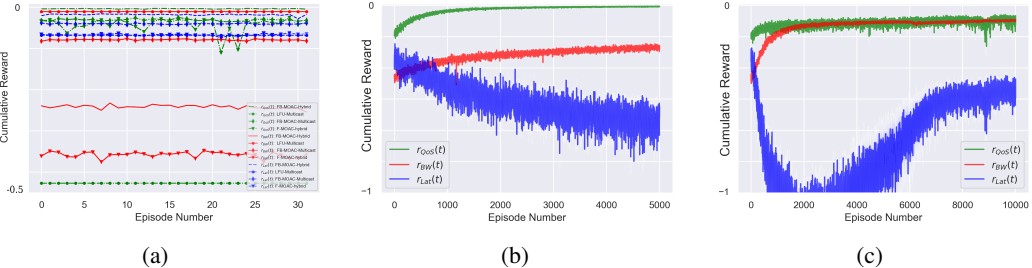

Figure 2: (a) Test result of FB-MOAC and random approaches on the hybrid and multicast schemes. Ablation of the (b) backward optimization and (c) the episodic MCS-average.

### 4.2.3 RESULTS

Figures 1b-1c illustrate the learning results of FB-MOAC algorithm for the hybrid and multicast environments as a function of episodes. The performance metrics are normalized so as to fit in one image. The results clearly shows that all of the considered performance metrics constantly improve, eventually evolving into a stable solution. They also highlight that the actor and forward/backward critic multi-objective agents are effectively learned.

We also consider two baseline schemes for comparison purposes: a widely-used rule-based approach; Least Frequently Used (LFU) strategy for the multicast experiment, and a learning-based algorithm for the hybrid experiment that replaces the backward MDP with a forward one. The LFU is widely utilized in the context of wireless caching (Ahmed et al., 2013). It keeps the most frequently requested contents in the caches of HNs. For the learning-based strategy, we consider this fact that optimizing the outage probability and controlling $d(t)$ reduce the expected latency based on Eq. (19). Hence, we consider the overall outage probability and bandwidth consumption as forward rewards, ignore the backward reward and control $d(t)$ to obtain a solution policy. Since the solution of this strategy can be obtained by a forward RL algorithm, we term this strategy as F-MOAC.

Figure 2a illustrates the test results of FB-MOAC algorithm and the baseline approaches; LFU and F-MOAC for the hybrid and multicast experiments as a function of episodes. Clearly, FB-MOAC solution outperforms the F-MOAC policy for the hybrid scheme in terms of all considered forward and backward rewards. Although the LFU strategy outperforms the FB-MOAC solution for the multicast experiment from latency and bandwidth perspectives, it cannot provide an acceptable solution since 80 percent of the requests will experience outage based on its QoS value, whereas only 1 percent of the requests will be lost for the FB-MOAC policy. These results indicate that the FB-MOAC algorithm can provide a stable and remarkably efficient solution.

### 4.2.4 ABLATION STUDY

We perform an ablation study to assess the benefit of the backward evaluation/optimization of FB-MOAC algorithm. For this, we disable the backward evaluation of the algorithm and only take into account the forward actor and critic losses. The study is to show that the backward reward does not increase simply as a result of optimizing the forward rewards, hence the objectives are conflicting. Figure 2b shows the learning result of FB-MOAC with only forward evaluation/optimization procedure for the hybrid experiments. It is apparent that the expected latency does not improve highlighting the importance of the backward evaluation/optimization procedure of FB-MOAC.

We perform an addition ablation study by disabling the *episodic MCS-average* mechanism of FB-MOAC. Figure 2c shows the learning performance of the hybrid experiment in this case. The result shows that the sample efficiency remarkable worsens when *episodic MCS-average* is not employed.

## 5 CONCLUSION

In this article, we developed a novel multi-objective RL algorithm, called FB-MOAC, for a class of forward-backward Markov decision process (FB-MDP) containing multiple tasks that conflict with each other. We then performed an convergence analysis on FB-MOAC algorithm under different assumptions. We evaluated FB-MOAC in multi-task experiments with FB-MDP environments. The extensive experiments and ablation study demonstrated the effectiveness of the solution of FB-

MOAC in deriving an optimal dynamic policy. Our work provides a novel mechanism of solving multi-task sequential-decision problems with controlled forward-backward dynamics.

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

SUPPLEMENTARY MATERIAL

## A  RELATED WORK

**Forward-Backward MDPs** Our paper has some similarity with prior research on RL algorithms (Edwards et al., 2018; Goyal et al., 2019; Wang et al., 2021; Archibald et al., 2023). These studies hypothesize that creating a virtual backward trajectory in relation to a forward MDP enhances the sample-efficiency of RL algorithms. Specifically, they employ the generated backward trajectories to augment the training dataset for learning forward MDP problems. Edwards et al. (2018) train a backwards dynamics to explore in a reverse direction from known goal states. The derived backward paths are then used to augment the replay buffer and contribute to the learning procedure of considered RL algorithm. Similarly, Goyal et al. (2019) learn an artificial backward model, called backtracking model, from the experiences performed by the agent interacting with the original forward dynamics. The backtracking model then enriches the training dataset by alternative trajectories leading a high value state. Lai et al. (2020); Wang et al. (2021) introduce learnable backward dynamics together with a novel reverse policy to generate paths towards the target states. In particular, they provide informed data augmentation for the training dataset by backpropagating through reverse paths Archibald et al. (2023) use the stochastic maximum principle, rather than dynamic programming framework, to obtain the optimum policy for a continuous RL problem with controlled forward dynamics or Forward Stochastic Differential Equation (FSDE). This approach results in a BSDE that needs to be jointly considered with the forward process in order to optimize the policy. Our model of interest is characterized based on both backward and forward rewards, in contrast with the studies (Edwards et al., 2018; Goyal et al., 2019; Lai et al., 2020; Wang et al., 2021) where backward dynamics are artificially constructed based on a forward MDP, and distinct from the approach discussed in (Archibald et al., 2023) where backward dynamics pertains to an FSDE problem. These rewards correspond to actual controlled backward and forward dynamics which compete with each other within the action space in both targeting directions of time. Consequently, our investigation is centered around a class of FB-MDPs of multi-task problems with conflicting forward and backward rewards.

**Stochastic policy RL algorithms** Here, we mention some relevant works (Qiu et al., 2021; Xu et al., 2020; Fu et al., 2021; Yang et al., 2018; Khodadadian et al., 2022) that explore the characterization of stochastic policy RL algorithms, such as Actor-Critic (AC) and Policy Gradient approaches (Sutton & Barto, 2018). Qiu et al. (2021) conduct a rigorous convergence analysis on the AC algorithm. Notably, their analysis is limited to a linear representation of the state-value function. Xu et al. (2020) provide a comprehensive characterization of the convergence rate and sample complexity of the Natural Actor-Critic (NAC) algorithm (Peters & Schaal, 2008). Their analysis relies on the transition probability of the considered MDP to be ergodic. Fu et al. (2021) analyze the convergence of the AC algorithm under the assumption that the considered family of Neural Networks (NNs) are closed under the Bellman operator. Lastly, Khodadadian et al. (2022) perform a meticulous convergence analysis of the Natural Policy Gradient algorithm (Kakade, 2002). However, their investigation assumes that the initialization value of the state-value function is sufficiently close to the optimal value function. All the aforementioned works address the convergence of stochastic policies of single-objective RL algorithms for forward MDP problems. In contrast, this work targets multi-task problems involving a FB-MDP. In this context, we carry out a rigorous convergence analysis as a solid foundation to characterize multi-objective and forward-backward RL algorithms.

## B  PROOF OF LEMMA 3.1

*Proof.* For the convenience, we indicate the conditional expectation $\mathbb{E}\{\cdot|\boldsymbol{\theta}\}$ simply by $\mathbb{E}\{\cdot\}$. We then have:

$$
\nabla_{\boldsymbol{\theta}} \mathbf{J}_b(\boldsymbol{\theta}) \stackrel{a}{=} \nabla_{\boldsymbol{\theta}} \, \mathbb{E}_{\mathrm{P}f(\boldsymbol{\tau})} \left\{ \mathbb{E}_{\mathrm{P}'_{\boldsymbol{\theta}}(\boldsymbol{\tau})} \left\{ \sum_{k=0}^{T-1} \gamma^k \boldsymbol{r}^b(\mathbf{y}_{T-k}, \mathbf{a}_{T-k}) \right\} \right\}
$$

$$
\stackrel{b}{=} \mathbb{E}_{\mathrm{P}f(\boldsymbol{\tau})} \left\{ \mathbb{E}_{\mathrm{P}'_{\boldsymbol{\theta}}(\boldsymbol{\tau})} \left\{ \sum_{k=0}^{T-1} \gamma^k \boldsymbol{r}^b(\mathbf{y}_{T-k}, \mathbf{a}_{T-k}) \sum_{k=0}^{T-1} \nabla_{\boldsymbol{\theta}} \log \pi_{\boldsymbol{\theta}}(\mathbf{a}_{T-k}|\mathbf{s}_{T-k}) \right\} \right\}
$$

$$
\stackrel{c}{=} \mathbb{E}_{\mathrm{P}f(\boldsymbol{\tau})} \left\{ \sum_{k=0}^{T-1} \mathbb{E}_{\mathrm{P}'_{\boldsymbol{\theta}}(\boldsymbol{\tau})} \left\{ \nabla_{\boldsymbol{\theta}} \log \pi_{\boldsymbol{\theta}}(\mathbf{a}_{T-k}|\mathbf{s}_{T-k}) \sum_{k'=k}^{T-1} \gamma^{k'-k} \boldsymbol{r}^b(\mathbf{y}_{T-k'}, \mathbf{a}_{T-k'}) \right\} \right\}
$$

$$
\stackrel{d}{=} \mathbb{E}_{\mathrm{P}f(\boldsymbol{\tau})} \left\{ \sum_{k=0}^{T-1} \mathop{\mathbb{E}}_{\substack{\mathbf{a}_{T-k} \sim \pi_{\boldsymbol{\theta}}(\cdot|\mathbf{s}_{T-k}) \\ \mathbf{y}_{T-k} \sim P_b(\cdot|\mathbf{y}_{T-k+1}, \mathbf{a}_{T-k+1})}} \left\{ \nabla_{\boldsymbol{\theta}} \log \pi_{\boldsymbol{\theta}}(\mathbf{a}_{T-k}|\mathbf{s}_{T-k}) \underbrace{\mathbb{E}\left\{ \sum_{k'=k}^{T-1} \gamma^{k'-k} \boldsymbol{r}^b(\mathbf{y}_{T-k'}, \mathbf{a}_{T-k'}) \big| \mathbf{y}_{T-k}, \mathbf{a}_{T-k} \right\}}_{Q^b(\mathbf{y}_{T-k}, \mathbf{a}_{T-k})} \right\} \right\}
$$

$$
\stackrel{e}{=} \mathbb{E}_{\mathrm{P}f(\boldsymbol{\tau})} \left\{ \sum_{k=0}^{T-1} \mathop{\mathbb{E}}_{\substack{\mathbf{a}_{T-k} \sim \pi_{\boldsymbol{\theta}}(\cdot|\mathbf{s}_{T-k}) \\ \mathbf{y}_{T-k} \sim P_b(\cdot|\mathbf{y}_{T-k+1}, \mathbf{a}_{T-k+1})}} \left\{ \nabla_{\boldsymbol{\theta}} \log \pi_{\boldsymbol{\theta}}(\mathbf{a}_{T-k}|\mathbf{s}_{T-k}) \Big( Q^b(\mathbf{y}_{T-k}, \mathbf{a}_{T-k}) - V^b(\mathbf{y}_{T-k}) \Big) \right\} \right\}
$$

$$
\stackrel{f}{=} \mathbb{E}_{\mathrm{P}f(\boldsymbol{\tau})} \left\{ \mathbb{E}_{\mathrm{P}'_{\boldsymbol{\theta}}(\boldsymbol{\tau})} \left\{ \sum_{k=0}^{T-1} \nabla_{\boldsymbol{\theta}} \log \pi_{\boldsymbol{\theta}}(\mathbf{a}_{T-k}|\mathbf{s}_{T-k}) \underbrace{\Big( \boldsymbol{r}^b(\mathbf{y}_{T-k}, \mathbf{a}_{T-k}) + \gamma V^b(\mathbf{y}_{T-k-1}) - V^b(\mathbf{y}_{T-k}) \Big)}_{:= \mathbf{A}^b(\mathbf{y}_{T-k}, \mathbf{a}_{T-k})} \right\} \right\}
$$

$$
= \mathbb{E}_{\mathrm{P}_{\boldsymbol{\theta}}(\boldsymbol{\tau})} \left\{ \sum_{k=0}^{T-1} \nabla_{\boldsymbol{\theta}} \log \pi_{\boldsymbol{\theta}}(\mathbf{a}_{T-k}|\mathbf{s}_{T-k}) \mathbf{A}^b(\mathbf{y}_{T-k}, \mathbf{a}_{T-k}) \right\}. \tag{23}
$$

where $V^b(\cdot) : \mathcal{Y} \to \mathbb{R}^{|S_b|}$, $Q^b(\cdot, \cdot) : \mathcal{Y} \times \mathcal{A} \to \mathbb{R}^{|S_b|}$ and $A^b(\cdot, \cdot) : \mathcal{Y} \times \mathcal{A} \to \mathbb{R}^{|S_b|}$ are the backward state-value, backward action-value, and backward advantage multivariate functions, respectively, and we have:

$$
V^b(\mathbf{y}_{T-k}) := \mathbb{E}\left\{ \sum_{k'=k}^{T-1} \gamma^{k'-k} \boldsymbol{r}^b(\mathbf{y}_{T-k'}, \mathbf{a}_{T-k'}) \big| \mathbf{y}_{T-k} \right\}. \tag{24}
$$

For (a), we regarded that the backward reward $\boldsymbol{r}^b(\cdot, \cdot)$ does not depend on forward states $\{\mathbf{s}_k\}_{k=1}^T$, for (b), we used $\nabla_{\boldsymbol{\theta}} \mathrm{P}'_{\boldsymbol{\theta}}(\boldsymbol{\tau}) = \mathrm{P}'_{\boldsymbol{\theta}}(\boldsymbol{\tau}) \nabla_{\boldsymbol{\theta}} \log \mathrm{P}'_{\boldsymbol{\theta}}(\boldsymbol{\tau})$, and $\nabla_{\boldsymbol{\theta}} \log \mathrm{P}'_{\boldsymbol{\theta}}(\boldsymbol{\tau}) = \sum_{k=0}^{T-1} \nabla_{\boldsymbol{\theta}} \log \pi_{\boldsymbol{\theta}}(\mathbf{a}_{T-k}|\mathbf{s}_{T-k})$ based on Eq. (3). For (c), we considered the anti-causality; the current action does not affect the future of backward rewards, for (d), the definition of action-value functions is applied, for (e), including a bias term, here $V^b(\mathbf{y}_{T-k})$, does not change the result due to $\mathbb{E}_{\mathbf{a}_{T-k} \sim \pi_{\boldsymbol{\theta}}(\cdot|\mathbf{s}_{T-k})} \{ \nabla_{\boldsymbol{\theta}} \log \pi_{\boldsymbol{\theta}}(\mathbf{a}_k|\mathbf{s}_k) \} = \mathbf{0}$ and for (f) we derive the Bellman's equation for the backward action-value function as follows:

$$
Q^b(\mathbf{y}_{T-k}, \mathbf{a}_{T-k})
$$

$$
= \int_{\substack{\{\mathbf{y}_{T-k}\}_k \\ \{\mathbf{a}_{T-k}\}_k}} \mathbb{E}\left\{ \sum_{k'=k}^{T-1} \gamma^{k'-k} \boldsymbol{r}^b(\mathbf{y}_{T-k'}, \mathbf{a}_{T-k'}) \big| \mathbf{y}_{T-k}, \mathbf{a}_{T-k}, \mathbf{y}_{T-k-1} \right\} P_b(\mathbf{y}_{T-k-1}|\mathbf{y}_{T-k}, \mathbf{a}_{T-k}) d\mathbf{y}_{T-k-1}
$$

$$
\stackrel{a}{=} \int \left( \boldsymbol{r}^b(\mathbf{y}_{T-k'}, \mathbf{a}_{T-k'}) + \gamma \mathop{\mathbb{E}}_{\substack{\{\mathbf{y}_{T-k}\}_k \\ \{\mathbf{a}_{T-k}\}_k}} \left\{ \sum_{k'=k+1}^{T-1} \gamma^{k'-k} \boldsymbol{r}^b(\mathbf{y}_{T-k'}, \mathbf{a}_{T-k'}) \big| \mathbf{y}_{T-k-1} \right\} \right) P_b(\mathbf{y}_{T-k-1}|\mathbf{y}_{T-k}, \mathbf{a}_{T-k}) d\mathbf{y}_{T-k-1}
$$

$$
= \mathop{\mathbb{E}}_{\mathbf{y}_{T-k-1} \sim P_b(\cdot|\mathbf{y}_{T-k}, \mathbf{a}_{T-k})} \left\{ \boldsymbol{r}^b(\mathbf{y}_{T-k}, \mathbf{a}_{T-k}) + \gamma V^b(\mathbf{y}_{T-k-1}) \right\} \tag{25}
$$

where for (a) we considered that $\mathbf{a}_{T-k}$ becomes independent from $\{\mathbf{a}_{T-k'-1}\}_{k'=k}^{T-1}$ as $\{\mathbf{s}_{T-k'-1}\}_{k'=k}^{T-1}$ are not included in the inner expectation. The same strategy can be done to obtain the Bellman's equation for the backward state-value function as follows:

$$
V^b(\mathbf{y}_{T-k}) = \mathop{\mathbb{E}}_{\substack{\mathbf{a}_{T-k} \sim \pi_{\boldsymbol{\theta}}(\cdot|\mathbf{s}_{T-k}) \\ \mathbf{y}_{T-k-1} \sim P_b(\cdot|\mathbf{y}_{T-k}, \mathbf{a}_{T-k})}} \left\{ \boldsymbol{r}^b(\mathbf{y}_{T-k}, \mathbf{a}_{T-k}) + \gamma V^b(\mathbf{y}_{T-k-1}) \, \big| \, \boldsymbol{\theta} \right\}.
$$

Note that no distinct forward and backward Bellman optimality equations do exist for the FB-MDPs. However, a Bellman Pareto-optimality equation can be instead found. For this, we consider this

fact that the forward and backward value functions become stationary when forward and backward transition probabilities are stationary. Then, by recalling the notion of Pareto-optimality and Pareto front, we define Pareto-optimal forward and backward value functions as follows:

$$\left[Q^{f^*}(\mathbf{s},\mathbf{a}), Q^{b^*}(\mathbf{y},\mathbf{a})\right] = \max_{\pi(\cdot|\cdot)} \left[Q^f(\mathbf{s},\mathbf{a}), Q^b(\mathbf{y},\mathbf{a})\right], \quad \left[V^{f^*}(\mathbf{s}), V^{b^*}(\mathbf{y})\right] = \max_{\pi(\cdot|\cdot)} \left[V^f(\mathbf{s}), V^b(\mathbf{y})\right],$$

for all $(\mathbf{s},\mathbf{y},\mathbf{a}) \in \mathcal{S} \times \mathcal{Y} \times \mathcal{A}$. Then, based on Eq. (25) and the Bellman's equation for the forward action-value function, we can get:

$$\left[Q^{f^*}(\mathbf{s},\mathbf{a}), Q^{b^*}(\mathbf{y},\mathbf{a})\right] = \left[\mathop{\mathbb{E}}_{\mathbf{s}^+ \sim P_f(\cdot|\mathbf{s},\mathbf{a})} \left\{\boldsymbol{r}^f(\mathbf{s},\mathbf{a}) + \gamma V^{f^*}\left(\mathbf{s}^+\right)\right\}, \mathop{\mathbb{E}}_{\mathbf{y}^- \sim P_b(\cdot|\mathbf{y},\mathbf{a})} \left\{\boldsymbol{r}^b(\mathbf{y},\mathbf{a}) + \gamma V^{b^*}\left(\mathbf{y}^-\right)\right\}\right],$$

where $\mathbf{y}^-$ is the backward state preceding $\mathbf{y}$ and $\mathbf{s}^+$ is the forward state following $\mathbf{s}$. By considering both forward and backward equations, we now obtain the following Bellman Pareto-optimality equation:

$$\left[V^{f^*}(\mathbf{s}), V^{b^*}(\mathbf{y})\right] = \max_{\mathbf{a}} \left[\mathop{\mathbb{E}}_{\mathbf{s}^+ \sim P_f(\cdot|\mathbf{s},\mathbf{a})} \left\{\boldsymbol{r}^f(\mathbf{s},\mathbf{a}) + \gamma V^{f^*}\left(\mathbf{s}^+\right)\right\}, \mathop{\mathbb{E}}_{\mathbf{y}^- \sim P_b(\cdot|\mathbf{y},\mathbf{a})} \left\{\boldsymbol{r}^b(\mathbf{y},\mathbf{a}) + \gamma V^{b^*}\left(y^-\right)\right\}\right],$$

This equation, termed as Bellman Pareto-optimality equation, provides a base to formulate forward-backward dynamic programming algorithms and motivates the usage of multi-objective optimization frameworks.

$\square$

## C  Convergence Analysis of FB-MOAC Algorithm

In this paper, we perform a comprehensive study of the convergence properties of the novel FB-MOAC algorithm. Our investigation commences with the establishment of some foundational assumptions and the introduction of preliminary theorems and corollaries. Subsequently, we study the convergence analysis for two distinct scenarios, specifically pertaining to the backward and forward expected losses: (1) **Strong Convexity and Lipschitz Smoothness Case.** We explore the convergence behavior when the losses exhibit both strong convexity and Lipschitz smoothness properties. (2) **Lipschitz Smoothness Case.** We investigate convergence when the losses are solely Lipschitz smooth. Through a rigorous examination of these cases, we thus intend to provide a comprehensive understanding of the convergence characteristics of FB-MOAC algorithm.

We also need to emphasize that stochastic nature of FB-MDP affects the values of $\phi$, $\psi$ and $\boldsymbol{\theta}$, based on the SGD rules (15) and (13), so they are treated as random variables.

We now make the following assumptions.

**Assumption 1:** The forward and backward state-value functions are unbiased:

$$\mathbb{E}_{\phi}\{V_{\phi,j}^f(\mathbf{s})\} = V_j^f(\mathbf{s}), \quad j \in S_f, \quad \mathbf{s} \in \mathcal{S}$$
$$\mathbb{E}_{\psi}\{V_{\psi,j}^b(\mathbf{y})\} = V_j^b(\mathbf{y}), \quad j \in S_b, \quad \mathbf{y} \in \mathcal{Y},$$

According to this assumption and Eqs. (14) and (17) we can get:

$$\nabla_{\boldsymbol{\theta}} \bar{J}_j^f(\boldsymbol{\theta}) = \mathbb{E}_{\phi} \mathbb{E}\left\{\nabla_{\boldsymbol{\theta}} \hat{J}_j^f(\boldsymbol{\theta},\phi) \mid \boldsymbol{\theta},\phi\right\} = -\sum_{k=1}^T \mathbb{E}_{\phi} \mathbb{E}\left\{\nabla_{\boldsymbol{\theta}} \log \pi_{\boldsymbol{\theta}}(\mathbf{a}_k|\mathbf{s}_k) A_{\phi,j}^f(\mathbf{s}_k,\mathbf{a}_k) \mid \boldsymbol{\theta}\right\}$$

$$= -\mathbb{E}\left\{\sum_{k=1}^T \nabla_{\boldsymbol{\theta}} \log \pi_{\boldsymbol{\theta}}(\mathbf{a}_k|\mathbf{s}_k) A_j^f(\mathbf{s}_k,\mathbf{a}_k) \mid \boldsymbol{\theta}\right\} = \nabla_{\boldsymbol{\theta}} J_j^f(\boldsymbol{\theta})$$

Likewise, it can be shown that:

$$\nabla_{\boldsymbol{\theta}} \bar{J}_j^b(\boldsymbol{\theta}) = -\mathbb{E}\left\{\sum_{k=1}^T \nabla_{\boldsymbol{\theta}} \log \pi_{\boldsymbol{\theta}}(\mathbf{a}_k|\mathbf{s}_k) A_j^b(\mathbf{y}_k,\mathbf{a}_k) \mid \boldsymbol{\theta}\right\} = \nabla_{\boldsymbol{\theta}} J_j^b(\boldsymbol{\theta})$$

**Assumption 2.1:** The forward and backward multi-objective expected losses are differentiable and strongly convex with parameters $\gamma_f$ and $\gamma_b$, respectively, w.r.t $\boldsymbol{\theta}$:

$$\nabla_{\boldsymbol{\theta}}^2 J_j^f(\boldsymbol{\theta}) - \gamma_f \boldsymbol{I} \succeq \boldsymbol{0}, \quad j \in S_f$$
$$\nabla_{\boldsymbol{\theta}}^2 J_j^b(\boldsymbol{\theta}) - \gamma_b \boldsymbol{I} \succeq \boldsymbol{0}, \quad j \in S_b.$$

**Assumption 2.2:** The forward and backward multi-objective expected losses are differentiable and Lipschitz smooth functions with constants $L_f$ and $L_b$, respectively, w.r.t $\boldsymbol{\theta}$:

$$\left\| \nabla_{\boldsymbol{\theta}} J_j^f(\boldsymbol{\theta}') - \nabla_{\boldsymbol{\theta}} J_j^f(\boldsymbol{\theta}) \right\| \leq L_f \|\boldsymbol{\theta}' - \boldsymbol{\theta}\|, \quad j \in S_f.$$
$$\left\| \nabla_{\boldsymbol{\theta}} J_j^b(\boldsymbol{\theta}') - \nabla_{\boldsymbol{\theta}} J_j^b(\boldsymbol{\theta}) \right\| \leq L_b \|\boldsymbol{\theta}' - \boldsymbol{\theta}\|, \quad j \in S_b.$$

Notice that Assumptions 2 is made for the expected losses ($J_j^f(\boldsymbol{\theta})$, $J_j^b(\boldsymbol{\theta})$) and not for the stochastic losses ($\hat{J}_j^f(\boldsymbol{\theta}, \boldsymbol{\phi})$, $\hat{J}_j^b(\boldsymbol{\theta}, \boldsymbol{\psi})$).

**Assumption 3:** Consider the following stochastic forward/backward gradient

$$\nabla \hat{\boldsymbol{J}}^{\text{fb}}(\boldsymbol{\theta}, \boldsymbol{\phi}, \boldsymbol{\psi}) = \left[ \left[ \nabla_{\boldsymbol{\theta}} \hat{J}_j^f(\boldsymbol{\theta}, \boldsymbol{\phi}) \right]_{j \in S_f}, \left[ \nabla_{\boldsymbol{\theta}} \hat{J}_j^b(\boldsymbol{\theta}, \boldsymbol{\psi}) \right]_{j \in S_b} \right],$$

then, its conditional covariance is bounded by a positive semi-definite matrix $\boldsymbol{B}$:

$$\mathbb{E} \left\{ \nabla \hat{\boldsymbol{J}}^{\text{fb}}(\boldsymbol{\theta}, \boldsymbol{\phi}, \boldsymbol{\psi})^\top \nabla \hat{\boldsymbol{J}}^{\text{fb}}(\boldsymbol{\theta}, \boldsymbol{\phi}, \boldsymbol{\psi}) \mid \boldsymbol{\theta} \right\} - \nabla \boldsymbol{J}^{\text{fb}}(\boldsymbol{\theta})^\top \nabla \boldsymbol{J}^{\text{fb}}(\boldsymbol{\theta}) \preceq \boldsymbol{B},$$

where

$$\nabla \boldsymbol{J}^{\text{fb}}(\boldsymbol{\theta})(\boldsymbol{\theta}, \boldsymbol{\phi}, \boldsymbol{\psi}) = \left[ \left[ \nabla_{\boldsymbol{\theta}} j^f(\boldsymbol{\theta}) \right]_{j \in S_f}, \left[ \nabla_{\boldsymbol{\theta}} J_j^b(\boldsymbol{\theta}) \right]_{j \in S_b} \right],$$

Note that the assumptions outlined in this context align with the conventions found in the literature related to the convergence analysis (Qiu et al., 2021; Zhou et al., 2022).

Now, we have the following theorem.

**Theorem C.1.** *Consider forward/backward expected losses, i.e., $\{J_j^f(\cdot)\}_{j \in S_f}$ and $\{J_j^b(\cdot)\}_{j \in S_b}$, and forward/backward stochastic losses, i.e., $\{\hat{J}_j^f(\cdot, \cdot)\}_{j \in S_f}$ and $\{\hat{J}_j^b(\cdot, \cdot)\}_{j \in S_b}$, complying with Assumptions 2.1, 2.2 and 3, and $\beta_{\text{act}}$ being the solution of Eq. (16). Moreover, consider SGDes (12) and (15) characterized by iteration number $i$ and actor learning rate $\{\mu_i\}_{i \in \mathcal{I}}$ with*

$$\mu_i \leq \min \left\{ \frac{1}{\max\{L_f, L_b\}}, \frac{1}{\max\{L_f, L_b\} \|\mathbf{B}\|} \mathbb{E} \left\{ \frac{1}{\mathbf{1}^\top \left( \nabla \boldsymbol{J}^{\text{fb}}(\boldsymbol{\theta}^i)^\top \nabla \boldsymbol{J}^{\text{fb}}(\boldsymbol{\theta}^i) \right)^{-1} \mathbf{1}} \right\} \right\},$$

*which generate sequences $\{\boldsymbol{\phi}^i\}_{i \in \mathcal{I}}$, $\{\boldsymbol{\psi}^i\}_{i \in \mathcal{I}}$ and $\{\boldsymbol{\theta}^i\}_{i \in \mathcal{I}}$, we then get:*

$$\mathbb{E} \, J_j^f(\boldsymbol{\theta}^{i+1}) \leq \mathbb{E} \, J_j^f(\boldsymbol{\theta}^i), \quad j \in S_f$$

*and*

$$\mathbb{E} \, J_j^b(\boldsymbol{\theta}^{i+1}) \leq \mathbb{E} \, J_j^b(\boldsymbol{\theta}^i), \quad j \in S_b.$$

*Proof.* According to the update rule Eq. (15), we have:

$$\boldsymbol{\theta}^{i+1} = \boldsymbol{\theta}^i - \mu_i \left[ \nabla \hat{\boldsymbol{J}}^{\text{f}}(\boldsymbol{\theta}^i, \boldsymbol{\phi}^i), \nabla \hat{\boldsymbol{J}}^{\text{b}}(\boldsymbol{\theta}^i, \boldsymbol{\psi}^i) \right] \beta_{\text{act}}^i = \boldsymbol{\theta}^i - \mu_i \nabla \hat{\boldsymbol{J}}^{\text{fb}}(\boldsymbol{\theta}^i, \boldsymbol{\phi}^i, \boldsymbol{\psi}^i) \beta_{\text{act}}^i.$$

Based on Assumption 2.2, we then obtain:

$$J_j^f(\boldsymbol{\theta}^{i+1}) - J_j^f(\boldsymbol{\theta}^i) \leq -\mu_i \nabla J_j^f(\boldsymbol{\theta}^i)^\top \nabla \hat{\boldsymbol{J}}^{\text{fb}}(\boldsymbol{\theta}^i, \boldsymbol{\phi}^i, \boldsymbol{\psi}^i) \beta_{\text{act}}^i + \frac{\mu_i^2 L_f}{2} \beta_{\text{act}}^{i}{}^\top \nabla \hat{\boldsymbol{J}}^{\text{fb}}(\boldsymbol{\theta}^i, \boldsymbol{\phi}^i, \boldsymbol{\psi}^i)^\top \nabla \hat{\boldsymbol{J}}^{\text{fb}}(\boldsymbol{\theta}^i, \boldsymbol{\phi}^i, \boldsymbol{\psi}^i) \beta_{\text{act}}^i.$$

By taking the expectation on both sides of the recent equation, it then reads:

$$\mathbb{E} \left\{ J_j^f(\boldsymbol{\theta}^{i+1}) - J_j^f(\boldsymbol{\theta}^i) \right\} \leq -\mu_i \mathbb{E} \left\{ \left( \boldsymbol{e}_j - \frac{\mu_i L_f}{2} \beta_{\text{act}}^i \right)^\top \nabla \boldsymbol{J}^{\text{fb}}(\boldsymbol{\theta}^i)^\top \nabla \boldsymbol{J}^{\text{fb}}(\boldsymbol{\theta}^i) \beta_{\text{act}}^i \right\} + \frac{\mu_i^2 L_f}{2} \|\boldsymbol{B}\|, \quad (26)$$

where was obtained based on

$$\mathbb{E}_{\boldsymbol{\phi}^i, \boldsymbol{\psi}^i} \mathbb{E} \left\{ \beta_{\text{act}}^{i}{}^\top \nabla \boldsymbol{J}^f(\boldsymbol{\theta}^i)^\top \nabla \hat{\boldsymbol{J}}^{\text{fb}}(\boldsymbol{\theta}^i, \boldsymbol{\phi}^i, \boldsymbol{\psi}^i) \beta_{\text{act}}^i \mid \boldsymbol{\theta}^i, \boldsymbol{\phi}^i, \boldsymbol{\psi}^i \right\} = \beta_{\text{act}}^{i}{}^\top \nabla \boldsymbol{J}^f(\boldsymbol{\theta}^i)^\top \nabla \boldsymbol{J}^{\text{fb}}(\boldsymbol{\theta}^i) \beta_{\text{act}}^i$$

according to Assumption 1, as well as based on Assumption 3 and $\boldsymbol{\beta}_{\text{act}}^i{}^\top \boldsymbol{B} \boldsymbol{\beta}_{\text{act}}^i \le \|\boldsymbol{B}\| \, \|\boldsymbol{\beta}_{\text{act}}^i\|^2 \le \|\boldsymbol{B}\|$. On the other hand, from Eq. (16), for all $\beta_{\text{act,j}}^i \ge 0$, it reads:

$$\boldsymbol{\beta}_{\text{act}}^i = \left(\mathbf{1}^\top \left(\nabla \boldsymbol{J}^{\text{fb}}(\boldsymbol{\theta}^i)^\top \nabla \boldsymbol{J}^{\text{fb}}(\boldsymbol{\theta}^i)\right)^{-1} \mathbf{1}\right)^{-1} \left(\nabla \boldsymbol{J}^{\text{fb}}(\boldsymbol{\theta}^i)^\top \nabla \boldsymbol{J}^{\text{fb}}(\boldsymbol{\theta}^i)\right)^{-1} \mathbf{1}.$$

By substituting this into Eq. (26), we get:

$$\mathbb{E}\left\{J_j^f(\boldsymbol{\theta}^{i+1}) - J_j^f(\boldsymbol{\theta}^i)\right\} \overset{a}{\le} -\frac{\mu_i}{2}\mathbb{E}\left\{\frac{1}{\mathbf{1}^\top \left(\nabla \boldsymbol{J}^{\text{fb}}(\boldsymbol{\theta}^i)^\top \nabla \boldsymbol{J}^{\text{fb}}(\boldsymbol{\theta}^i)\right)^{-1} \mathbf{1}}\right\} + \frac{\mu_i^2 L_f}{2}\|\boldsymbol{B}\| \ \le \ 0,$$

where we used $\mu_i \max\{L_f, L_b\} \le 1$ for (a). Considering that the denominator of RHS of the recent equation is positive due to the positive-definiteness of $\left(\nabla \boldsymbol{J}^{\text{fb}}(\boldsymbol{\theta}^i)^\top \nabla \boldsymbol{J}^{\text{fb}}(\boldsymbol{\theta}^i)\right)^{-1}$, the statement follows. The same analysis can be done to infer $\mathbb{E}\left\{J_j^b(\boldsymbol{\theta}^{i+1}) - J_j^b(\boldsymbol{\theta}^i)\right\} \le 0$ ☐

***Remarks.*** Theorem C.1 guarantees all the forward and backward expected losses $\{\mathbb{E} \, J_j^f(\boldsymbol{\theta})\}_{j \in S_f}$ and $\{\mathbb{E} \, J_j^b(\boldsymbol{\theta})\}_{j \in S_b}$ continually reduce as the algorithm iteration increases. It thus enables us to jointly improve all of the cumulative rewards, either forward or backward, with each iteration, on average.

**Corollary C.1.** *Consider the framework of Lemma C.1, we then get:*

$$\mathbb{E}\left\{\boldsymbol{\beta}_{\text{act}}^i{}^\top \nabla \boldsymbol{J}^{\text{fb}}(\boldsymbol{\theta}^i)^\top \nabla \boldsymbol{J}^{\text{fb}}(\boldsymbol{\theta}^i) \boldsymbol{\beta}_{\text{act}}^i\right\} \le \frac{2}{\mu_i}\mathbb{E}\left\{\sum_{j \in S_f \cup S_b} \beta_{\text{act,j}}^i \left(J_j^{\text{fb}}(\boldsymbol{\theta}^i) - J_j^{\text{fb}}(\boldsymbol{\theta}^{i+1})\right)\right\} + \mu_i \max\{L_f, L_b\}\|\boldsymbol{B}\|.$$

*Proof.* Based on Eq. (26) and $\mu_i \max\{L_f, L_b\} \le 1$, the statement follows. ☐

## C.1 CONVERGENCE ANALYSIS FOR THE CASE OF STRONGLY-CONVEX AND SMOOTHNESS

**Theorem C.2.** *Consider the framework of Theorem C.1, and assume forward-backward multi-objective optimization $O_2$ with a $\boldsymbol{\theta}$-parametric policy distribution $\pi_{\boldsymbol{\theta}}(\cdot|\cdot)$ being optimized by SGDs (15) and (13) with generated sequences $\{\boldsymbol{\theta}^i\}_{i \in \mathcal{I}}$, $\{\boldsymbol{\phi}^i\}_{i \in \mathcal{I}}$ and $\{\boldsymbol{\psi}^i\}_{i \in \mathcal{I}}$, and actor learning rate $\{\mu_i\}_{i \in \mathcal{I}}$ complying with assumptions of Theorem C.1. Furthermore, assume there exists a Pareto optimal solution $\boldsymbol{\theta}^*$ of $O_2$ dominating $\boldsymbol{\theta}^i$ for the objectives $\{J_j^{\text{fb}}(\cdot)\}_{j \in S_f \cup S_b}$ with $i \in \mathcal{I}$. Then, we have:*

$$\mathbb{E}\|\boldsymbol{\theta}^{i+1} - \boldsymbol{\theta}^*\| \le \left(1 - \max\{\gamma_f, \gamma_b\}\mu_i\right) \mathbb{E}\|\boldsymbol{\theta}^i - \boldsymbol{\theta}^*\| + \left(1 + \max\{L_f, L_b\}\right)\mu_i^2\|\boldsymbol{B}\|. \tag{27}$$

*Proof.* Based on SGD update (15), we obtain:

$$\mathbb{E}\|\boldsymbol{\theta}^{i+1} - \boldsymbol{\theta}^*\|^2 = \mathbb{E}\|\boldsymbol{\theta}^i - \boldsymbol{\theta}^* - \mu_i \nabla \hat{\boldsymbol{J}}^{\text{fb}}(\boldsymbol{\theta}^i, \boldsymbol{\phi}^i, \boldsymbol{\psi}^i)\boldsymbol{\beta}_{\text{act}}^i\|^2 \le \mathbb{E}\|\boldsymbol{\theta}^i - \boldsymbol{\theta}^*\|^2$$

$$- 2\mu_i \mathbb{E}\left\{\mathbb{E}\left\{\sum_{j \in S_f \cup S_b} \beta_{\text{act,j}}^i \nabla \hat{J}_j^{\text{fb}}(\boldsymbol{\theta}^i, \boldsymbol{\phi}^i, \boldsymbol{\psi}^i)^\top \left(\boldsymbol{\theta}^i - \boldsymbol{\theta}^*\right) \mid \boldsymbol{\theta}^i, \boldsymbol{\phi}^i, \boldsymbol{\psi}^i\right\}\right\}$$

$$+ \mathbb{E}\left\{\mu_i^2 \boldsymbol{\beta}_{\text{act}}^i{}^\top \nabla \hat{\boldsymbol{J}}^{\text{fb}}(\boldsymbol{\theta}^i, \boldsymbol{\phi}^i, \boldsymbol{\psi}^i)^\top \nabla \hat{\boldsymbol{J}}^{\text{fb}}(\boldsymbol{\theta}^i, \boldsymbol{\phi}^i, \boldsymbol{\psi}^i)\boldsymbol{\beta}_{\text{act}}^i\right\}$$

$$\overset{a}{\le} \mathbb{E}\|\boldsymbol{\theta}^i - \boldsymbol{\theta}^*\|^2 - 2\mu_i \mathbb{E}\left\{\sum_{j \in S_f \cup S_b} \beta_{\text{act,j}}^i \nabla J_j^{\text{fb}}(\boldsymbol{\theta}^i)^\top \left(\boldsymbol{\theta}^i - \boldsymbol{\theta}^*\right)\right\}$$

$$+ \mathbb{E}\left\{\mu_i^2 \boldsymbol{\beta}_{\text{act}}^i{}^\top \nabla \hat{\boldsymbol{J}}^{\text{fb}}(\boldsymbol{\theta}^i, \boldsymbol{\phi}^i, \boldsymbol{\psi}^i)^\top \nabla \hat{\boldsymbol{J}}^{\text{fb}}(\boldsymbol{\theta}^i, \boldsymbol{\phi}^i, \boldsymbol{\psi}^i)\boldsymbol{\beta}_{\text{act}}^i\right\}$$

$$\overset{b}{\le} \left(1 - \max\{\gamma_f, \gamma_b\}\mu_i\right) \mathbb{E}\|\boldsymbol{\theta}^i - \boldsymbol{\theta}^*\|^2 + 2\mu_i \mathbb{E}\left\{\sum_{j \in S_f \cup S_b} \beta_{\text{act,j}}^i \left(J_j^{\text{fb}}(\boldsymbol{\theta}^*) - J_j^{\text{fb}}(\boldsymbol{\theta}^i)\right)\right\}$$

$$+ \mu_i^2 \mathbb{E}\left\{\mathbb{E}\left\{\boldsymbol{\beta}_{\text{act}}^i{}^\top \nabla \hat{\boldsymbol{J}}^{\text{fb}}(\boldsymbol{\theta}^i, \boldsymbol{\phi}^i, \boldsymbol{\psi}^i)^\top \nabla \hat{\boldsymbol{J}}^{\text{fb}}(\boldsymbol{\theta}^i, \boldsymbol{\phi}^i, \boldsymbol{\psi}^i)\boldsymbol{\beta}_{\text{act}}^i \mid \boldsymbol{\theta}^i, \boldsymbol{\phi}^i, \boldsymbol{\psi}^i\right\}\right\}$$

$$\overset{c}{\le} \left(1 - \max\{\gamma_f, \gamma_b\}\mu_i\right)\mathbb{E}\|\boldsymbol{\theta}^i - \boldsymbol{\theta}^*\|^2 + \mu_i^2\|\boldsymbol{B}\| + 2\mu_i \mathbb{E}\left\{\sum_{j \in S_f \cup S_b} \beta_{\text{act,j}}^i \left(J_j^{\text{fb}}(\boldsymbol{\theta}^*) - J_j^{\text{fb}}(\boldsymbol{\theta}^{i+1})\right)\right\}$$

$$\overset{d}{\le} \left(1 - \max\{\gamma_f, \gamma_b\}\mu_i\right) \mathbb{E}\|\boldsymbol{\theta}^i - \boldsymbol{\theta}^*\|^2 + \left(1 + \max\{L_f, L_b\}\right)\mu_i^2\|\boldsymbol{B}\|,$$

where (a) was achieved considering $\mathbb{E}_{\phi,\psi}\mathbb{E}\{\nabla\hat{\boldsymbol{J}}^{\mathrm{fb}}(\boldsymbol{\theta},\phi,\psi)|\boldsymbol{\theta},\phi,\psi\} = \nabla\boldsymbol{J}^{\mathrm{fb}}(\boldsymbol{\theta})$ based on Assumption 1, (b) was obtained based on Assumption 2.1, (c) according to Assumption 3 and Corollary C.1, and for (d) we exploited $\boldsymbol{\theta}^*$ being a dominating Pareto optimum. $\qquad\square$

Accordingly, we can get:

**Corollary C.2.** *Consider the SGD approach (15) with generated sequence $\{\boldsymbol{\theta}^n\}_{n\in\mathcal{I}}$ and actor learning rate $\{\mu_n\}_{n\in\mathcal{I}}$ complying with assumptions of Theorem C.1. Set $\mu_n$ so that $\lim_{n\to\infty}\mu_n = 0$, then $\lim_{n\to\infty}\mathbb{E}\|\boldsymbol{\theta}^n - \boldsymbol{\theta}^*\|^2 = 0$.*

*Proof.* We use the result provided in Turinici (2021). As such, based on Theorem C.2 we have:

$$\begin{aligned}\Delta_{n+1} - \epsilon &\leq \big(1 - \max\{\gamma_f,\gamma_b\}\mu_n\big)(\Delta_n - \epsilon) - \mu_n\big(\max\{\gamma_f,\gamma_b\}\epsilon - (1+\max\{L_f,L_b\})\mu_n\|\boldsymbol{B}\|\big)\\ &\overset{a}{\leq} \big(1 - \max\{\gamma_f,\gamma_b\}\mu_n\big)(\Delta_n - \epsilon).\end{aligned}$$

where $\Delta_n = \mathbb{E}\|\boldsymbol{\theta}^n - \boldsymbol{\theta}^*\|^2$ and $\epsilon > 0$. For (a), we considered that $\max\{\gamma_f,\gamma_b\}\epsilon - (1+\max\{L_f,L_b\})\mu_n\|\boldsymbol{B}\| \geq 0$ for large $n$. Hence, for $\max\{\gamma_f,\gamma_b\}\mu_n \leq 1$ it reads:

$$[\Delta_{n+1} - \epsilon]^+ \leq \big(1 - \max\{\gamma_f,\gamma_b\}\mu_n\big)[\Delta_n - \epsilon]^+,$$

where $[x]^+ = x + |x|$. By iterating, we get:

$$[\Delta_{n+k} - \epsilon]^+ \leq \prod_{i=0}^{k-1}\big(1 - \max\{\gamma_f,\gamma_b\}\mu_{n+i}\big)[\Delta_n - \epsilon]^+.$$

Considering that $\lim_{k\to\infty}\prod_{i=0}^{k-1}\big(1 - \max\{\gamma_f,\gamma_b\}\mu_{n+i}\big) = 0$, we have: $\lim_{m\to\infty}[\Delta_m - \epsilon]^+$. Since it holds for any value $\epsilon > 0$, the statement follows. $\qquad\square$

***Remarks.*** The results of Theorem C.2 and Corollary C.2 guarantees that a convergence-in-mean towards a Pareto optimal solution can be achieved by choosing a proper actor learning rate. More specifically, a convergence-in-mean with the rate of $\mathcal{O}(1/|\mathcal{I}|)$ can be achieved for the learning rate being set to $\mu_i = \mathcal{O}(1/i)$.

***Remarks.*** The result of Corollary C.2 can be verified in a distinct quantitative way. More specifically, for the learning rate $\mu_i$ being sufficiently small, the evolution of SGD equation 15 can be regarded in a continuous time flow with parameter $t$. As such, based on 27, the dynamics of $\Delta_i = \mathbb{E}\|\boldsymbol{\theta}^i - \boldsymbol{\theta}^*\|$ can be expressed by the following inequality:

$$\Delta_\tau \leq \Delta_0\,\exp\Big(-\max\{\gamma_f,\gamma_b\}\int_0^\tau \mu_s ds\Big) + \big(1+\max\{L_f,L_b\}\big)\|\boldsymbol{B}\|\int_0^\tau \mu_t^2\,\exp\Big(-\max\{\gamma_f,\gamma_b\}\int_t^\tau \mu_s ds\Big)dt.$$

Then, it can be confirmed that for the selection $\mu_t = \frac{c_0}{t}$, we can get $\lim_{\tau\to\infty}\Delta_\tau \to 0$ yielding:

$$\lim_{i\to\infty}\mathbb{E}\|\boldsymbol{\theta}^i - \boldsymbol{\theta}^*\| \to 0.$$

## C.2 CONVERGENCE ANALYSIS FOR THE CASE OF SMOOTHNESS

Assuming strong-convexity for the expected losses may not be a reasonable assumption, particularly when the NN architecture of the actor agent exhibits non-linearity. Motivated by this fact, we perform a convergence analysis without imposing the strong-convexity assumption, focusing solely on the smoothness condition as defined in Assumption 2.2. In light of this approach, we thus have the following theorem.

**Theorem C.3.** *Consider forward/backward expected losses, i.e., $\{J_j^f(\cdot)\}_{j\in S_f}$ and $\{J_j^b(\cdot)\}_{j\in S_b}$, and forward/backward stochastic losses, i.e., $\{\hat{J}_j^f(\cdot,\cdot)\}_{j\in S_f}$ and $\{\hat{J}_j^b(\cdot,\cdot)\}_{j\in S_b}$, complying with Assumptions 2.2 and 4, and $\beta_{\mathrm{act}}$ being the solution of Eq. (16). Moreover, consider SGDs (12) and (15) characterized by iteration number $i$ and actor learning rate $\{\mu_i\}_{i\in\mathcal{I}}$ with*

$$\mu_i \leq \min\left\{\frac{1}{\max\{L_f,L_b\}}, \frac{1}{\max\{L_f,L_b\}\|\boldsymbol{B}\|}\left(\mathbf{1}^\top\big(\nabla\boldsymbol{J}^{\mathrm{fb}}(\boldsymbol{\theta}^i)^\top\nabla\boldsymbol{J}^{\mathrm{fb}}(\boldsymbol{\theta}^i)\big)^{-1}\mathbf{1}\right)^{-1}\right\},$$

*and $0 < \mu_{|\mathcal{I}|} \leq \ldots \leq \mu_i \leq \ldots \leq \mu_1$, which generate sequences $\{\phi^i\}_{i\in\mathcal{I}}$, $\{\psi^i\}_{i\in\mathcal{I}}$ and $\{\theta^i\}_{i\in\mathcal{I}}$. Then, we get:*

$$\min_{i\in\mathcal{I}} \mathbb{E}\Big\{ \|\nabla \boldsymbol{J}^{\mathrm{fb}}(\boldsymbol{\theta}^i)\boldsymbol{\beta}_{\mathrm{act}}^i\|^2 \Big\} \leq \frac{\max\{L_f, L_b\}\,\|\boldsymbol{B}\|}{|\mathcal{I}|} \sum_{i\in\mathcal{I}} \frac{\mu_i}{2 - \mu_i \max\{L_f, L_b\}}$$

$$+ \frac{2}{|\mathcal{I}|\,\mu_{|\mathcal{I}|}} \sum_{j\in S_f\cup S_b} \mathbb{E}\big\{ J_j^{\mathrm{fb}}(\boldsymbol{\theta}^1) - J_j^{\mathrm{fb}}(\boldsymbol{\theta}^{|\mathcal{I}|}) \big\}. \tag{28}$$

*where $\boldsymbol{J}^{\mathrm{fb}}(\boldsymbol{\theta}) = \Big[ [J_j^f(\boldsymbol{\theta})]_{j\in S_f}, [J_j^b(\boldsymbol{\theta})]_{j\in S_b} \Big].$*

*Proof.* Based on Eq. (26), we can get:

$$\sum_{j\in S_f\cup S_b} \beta_{\mathrm{ac},j}^i \big( J_j^{\mathrm{fb}}(\boldsymbol{\theta}^{i+1}) - J_j^{\mathrm{fb}}(\boldsymbol{\theta}^i) \big) \leq \mu_i \big( \frac{\mu_i}{2} \max\{L_f, L_b\} - 1 \big) \boldsymbol{\beta}_{\mathrm{act}}^i{}^\top \nabla \boldsymbol{J}^{\mathrm{fb}}(\boldsymbol{\theta}^i)^\top \nabla \boldsymbol{J}^{\mathrm{fb}}(\boldsymbol{\theta}^i)\boldsymbol{\beta}_{\mathrm{act}}^i$$

$$+ \frac{\mu_i^2}{2} \max\{L_f, L_b\}\|\boldsymbol{B}\|.$$

Therefore, we have:

$$\mathbb{E}\Big\{ \|\nabla \boldsymbol{J}^{\mathrm{fb}}(\boldsymbol{\theta}^i)\boldsymbol{\beta}_{\mathrm{ac}}^i\|^2 \Big\} \leq \frac{1}{\mu_i(1 - \frac{\mu_i}{2} \max\{L_f, L_b\})} \mathbb{E}\Big\{ \sum_{j\in S_f\cup S_b} \beta_{\mathrm{act},j}^i \big( J_j^{\mathrm{fb}}(\boldsymbol{\theta}^i) - J_j^{\mathrm{fb}}(\boldsymbol{\theta}^{i+1}) \big) \Big\}$$

$$+ \frac{\mu_i \max\{L_f, L_b\}}{2 - \mu_i \max\{L_f, L_b\}}\|\boldsymbol{B}\|.$$

By using the result of Theorem C.1, $\beta_{\mathrm{act},j} \leq 1$ for $j \in S_f \cup S_b$, and applying telescopic cancellation, the statement follows. $\qquad\square$

***Remarks.*** Note that the result of Theorem C.3 implies the converges to a locally Pareto optimal solution (Zhou et al., 2022) provided a suitable dynamics for the learning rate is chosen. Specifically, it suggests a convergence with the rate of $\mathcal{O}(1/\sqrt{|\mathcal{I}|})$ for the learning rate being set to $\mu_i = \mathcal{O}(1/\sqrt{i})$.

***Remarks.*** Notice that the selection $\mu_i = \mathcal{O}(1/i)$ cannot lead to a convergence due to the second term in the RHS of Eq. (28). This is contrast to the findings in Section C.1, where a convergence rate of $\mathcal{O}(1/|\mathcal{I}|)$ can be achieved for the strongly-convex and Lipschitz smooth case.

## D  FB-MOAC ALGORITHM

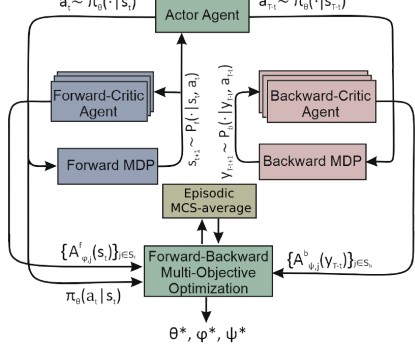

Figure 3: Illustration of the proposed FB-MOAC algorithm.

---

**Algorithm 1:** Pseudo-Code of *FB-MOAC*.

---

**for** $episode = 1$ **to** $E_{\max}$ **do**

 **Input:** Initial forward-backward states $(\mathbf{s}_1, \mathbf{y}_t)$, actor, forward-critic and backward-critic agents parameterized by $\boldsymbol{\theta}$, $\boldsymbol{\phi}$ and $\boldsymbol{\psi}$.

 **Forward Evaluation:**

 **for** $t = 1$ **to** $T$ **do**

  Select action $\mathbf{a}_t$ following $\pi_{\boldsymbol{\theta}}(\cdot|\mathbf{s}_t)$, interact with the environment.

  Observe new forward state $\mathbf{s}_{t+1}$ and forward immediate rewards $\{r_j^f(\mathbf{s}_t, \mathbf{a}_t)\}_{j \in S_f}$.

  Computes $\{A_{\boldsymbol{\phi},j}^f(\mathbf{s}_t, \mathbf{a}_t)\}_{j \in S_f}$ using forward state-value functions $\{V_{\boldsymbol{\phi},j}^f(\mathbf{s}_t)\}_{j \in S_f}$ using Eq. (9).

  Compute $\log(\pi_{\boldsymbol{\theta}}(\mathbf{a}_t|\mathbf{s}_t))$.

 **end**

 **Backward Evaluation:**

 **for** $t = 1$ **to** $T$ **do**

  Based on the chosen action of step *Forward-Evaluation*, $\mathbf{a}_{T-t}$, observe new backward state $\mathbf{y}_{T-t}$ and backward immediate rewards $\{r_j^b(\mathbf{y}_{T-t}, \mathbf{a}_{T-t})\}_{j \in S_b}$.

  Computes $\{A_{\boldsymbol{\psi},j}^b(\mathbf{y}_{T-t}, \mathbf{a}_{T-t})\}_{j \in S_b}$ using backward state-value functions $\{V_{\boldsymbol{\psi},j}^b(\mathbf{y}_{T-t})\}_{j \in S_b}$ based on Eq. (10).

 **end**

 **Forward-Backward Optimization:**

  **Forward/Backward Critic Update**:

  Obtain $\boldsymbol{\beta}_f^*$ and $\boldsymbol{\beta}_b^*$ based on Eq. (12).

  Compute multi-objective forward-critic loss $K^f(\boldsymbol{\phi})$ and backward-critic loss $K^b(\boldsymbol{\psi})$, and apply the rules:

$$\boldsymbol{\phi} \leftarrow \boldsymbol{\phi} - \mu_f \nabla_{\boldsymbol{\phi}} K^f(\boldsymbol{\phi}), \qquad \boldsymbol{\psi} \leftarrow \boldsymbol{\psi} - \mu_b \nabla_{\boldsymbol{\psi}} K^b(\boldsymbol{\psi}).$$

  **Forward-Backward Optimization**:

  Obtain $\boldsymbol{\beta}^*$ based on Eq. (16) and the outcomes of *episodic MCS-average*.

  Compute stochastic forward and backward gradients $\nabla_{\boldsymbol{\theta}} \hat{J}_j^f(\boldsymbol{\theta}, \boldsymbol{\phi})$ and $\nabla_{\boldsymbol{\theta}} \hat{J}_j^b(\boldsymbol{\theta}, \boldsymbol{\psi})$ using Eq. (14). Apply the SGD rule:

$$\boldsymbol{\theta} \leftarrow \boldsymbol{\theta} - \mu \Big( \sum_{j \in S_f} \beta_{\text{act,j}} \nabla_{\boldsymbol{\theta}} \hat{J}_j^f(\boldsymbol{\theta}, \boldsymbol{\phi}) + \sum_{j \in S_b} \beta_{\text{act,j}} \nabla_{\boldsymbol{\theta}} \hat{J}_j^b(\boldsymbol{\theta}, \boldsymbol{\psi}) \Big),$$

**end**

---

# E MOTIVATING EXAMPLES OF FB-MDPS

**Example 1**: The first example is in the domain of computation offloading (Zabihi et al., 2023). Let us consider a resource-constrained mobile device that needs to carry out computational intensive tasks. Instead of running the tasks locally, the mobile devices instead *offloads* them to an edge server, which processes them according to its computational capacity. The edge server has a buffer to store the tasks that cannot be immediately processed, which happens when there is no spare computational capacity. Assume that the average processing time of a buffered task at time $t$ is equal to $\mathrm{d}(t)$, and the buffer overflows with probability $\mathrm{O}(t)$. The aim is to calculate the average time $\mathrm{T}(t)$ for an offloaded task to be successfully computed. By the law of total expectation, we can thus obtain:

$$T(t) = O(t)(\tau + T(t+1)) + (1 - O(t))d(t),$$

where $\tau$ is the transmission time to prepare and re-offload the task. Therefore, the experienced computation time depends on its value in future and the system parameters $\mathrm{O}(t)$ and $\mathrm{d}(t)$. This thus portrays a backward MDP as the parameters $\mathrm{O}(t)$ and $\mathrm{d}(t)$ depend on the system action parameters, including, the buffer size, the computation capacity, and the task priority.

**Example 2**: The second example is in the context of cache-aided transmission schemes (Nomikos et al., 2022). It can be shown that for error-prone transmission approaches which operate in time-slotted fashion and serves users by N different contents, the environment leads to a controlled forward-backward dynamics. By denoting the error probability for reception of file $n$ at time-slot $t$ with $O_n(t)$, and the content popularity of file $n$ with $q_n(t)$, the request probability of file $n$ can be

found by:

$$P_n(t) = q_n(t) \sum_{n=1}^{N} p_n(t-1)\left(1 - O_n(t)\right) + p_n(t-1)O_n(t).$$

This was obtained by assuming this fact that a user repeats its request if it cannot successfully receive the file. Note that this equation leads to a controlled forward dynamics as the error probability depends on system action parameters. Moreover, the average latency $L_n(t)$ experienced by a typical user to successfully receive file $n$ can be expressed by:

$$L_n(t) = d_n(t)\left(1 - O_n(t)\right) + \left(\tau(t) + L_n(t+1)\right)O_n(t).$$

where $d_n(t)$ is the average reception delay for file $n$ if it is successfully served, and $\tau(t)$ is duration of time-slot $t$. This equation leads to a controlled backward dynamics as the parameter $d_n(t)$ and $O_n(t)$ depend on the system action parameters.

