# OpenReview forum: "Multi-Objective Reinforcement Learning for Forward-Backward Markov Decision Processes"
_ICLR.cc/2024/Conference — Submitted to ICLR 2024_

### Official Review · Reviewer_EeKJ · 2023-10-31

**Soundness:** 3 good
**Presentation:** 2 fair
**Contribution:** 2 fair
**Rating:** 5
**Confidence:** 4

**Summary:**

In this paper, the authors have devised a new approach known as Forward Backward Multi-objective Reinforcement Learning (FB-MORL). The convergence behavior of the proposed approach to Pareto optimality is analyzed. The effectiveness of the proposed approach is evaluated on a use-case from wireless caching.

**Strengths:**

A new approach is proposed for FB-MORL. A real-world wireless communication problem has been taken as a use case of the proposed methodology.

**Weaknesses:**

The presentation of the paper needs to be improved. A small real-world example illustrating FBMDP would help strengthen the motivation. The algorithm may be computationally expensive.

**Questions:**

My comments are as follows:

1.	How do the forward and backward processes conflict with each other through the action space? A small real-world example illustrating this would help strengthen the motivation.

2.	The authors stated that the backward dynamics in the backward Markov Decision Process (MDP)  result in a known final state. Can’t it be analyzed by a forward MDP with an absorbing state?

3.	In Section 3.2, it is considered that the backward and forward trajectories conflict with each other. However, the analysis does not reflect that. Rather, it seems the analysis presented in the paper replicates the existing forward MDP results for backward MDP. For example, Lemma 4.1 is just an extension of (5). What are the additional challenges in obtaining the convergence results?

4.	In Equation (2), why is $\gamma \in (0,1]$? Usually for a finite $T$, $\gamma=1$ is more appropriate.

5.	In Equation (3), why is the $\pi_\theta$ conditioned on $s_t$ and not on $y_t$?

6.	The computation of  $\beta_f^*$ and $\beta_b^*$ needs to be done at every iteration. Is it computationally feasible?

7.	Can’t equation (18) be alternatively expressed as a forward MDP? In other words, can the problem in 5.2.1 be modeled with forward-only dynamics?

8.	The presentation of the paper needs to be improved. The  FB-MOAC algorithm should be moved from the appendix to the main text since this is one of the main contributions of the paper. What are the additional challenges in obtaining the analytical results in Section 5.1 compared to the forward MDP only case?

---

> ### Author Response · Authors · 2023-11-17
>
> “_example illustrating FBMDP strengthen the motivation. algorithm may be computationally expensive_”
>
> We have included an example in the context of computation offloading in the revised submission to motivate the applicability of FB-MDPs to a different scenario. Computational complexity is determined by two factors: the selection of the learning rate and the episodic MCS-average mechanism. The former is not a concern, as the simulations suggest that even a fixed learning rate obtains convergence. The latter is primarily affected by the number $N_{MCS}$ of critic agents that should be implemented. Based on simulations, 3 or 4 are sufficient to reach convergence. As a consequence, the complexity of FB-MOAC is competitive with respect to a forward-only RL algorithm.
>
> “_How do forward and backward processes conflict_”
>
> The forward and backward dynamics compete with each other as they depend on the action variables and they correspond to action-dependent rewards. Specifically, the optimal action from the perspective of forward rewards [Eqs. (20) and (21)] are not necessarily aligned with the optimal action from the perspective of backward reward [Eq. (22)]. This has been numerically evaluated by the ablation study in Section 4.2.3 of the revised manuscript. In particular, Figure 2 shows that merely optimizing the forward rewards causes the backward reward to diverge.
>
> “_Can’t it be analyzed by a forward MDP with an absorbing state_”
>
> The backward dynamics starts from a known state, however, it moves backwards over time: the current state depends on the future state and action, in contrast with the forward dynamics. Note that converting the backward dynamics by flipping the time does not solve the issue as the resulting state would now depend on the actions that happen in a far future. Hence, it cannot actually be formulated according to a forward-MDP with absorbing states.
>
> “_The analysis does not reflect that conflict of trajectories_”
>
> We appreciate this remark. Lemma 3.1 differs from its counterpart for the forward dynamics as: it depends on the backward reward; the transition trajectory and the backward value function in Eq. (7) are obtained based on a policy distribution that only depends on the forward state. Hence, they are in conflict since the rewards compete, resulting in several challenges: jointly optimizing the action-coupled backward and forward rewards; devising a multi-objective algorithm that updates the policy _after_ the forward and backward dynamics are evaluated; and the convergence analysis for the devised multi-objective actor agent, shared between the forward and backward dynamics.
>
> “_why is γ ∈(0,1] ?_”
>
> In typical problems, setting $\gamma=1$ generally suffices. In our case, however, we observed that the backward reward diverges for $\gamma>0.97$ and the sample-efficiency of the algorithm remarkably worsens for $\gamma<0.9$.
>
>
> “_why is the \pi(.) conditioned on s_t and not on y_t_”
>
> The backward state moves backwards over time and is not known in advance. The strategy $\pi(. | s_t)$ simplifies the algorithm derivation. Moreover, it does not constrain solution optimality since it is updated after both the forward and backward procedures are evaluated, and it drives the backward state to _optimally_ evolve based on this optimal policy. Hence, both the backward and forward rewards are jointly optimized. Note that a policy equal to $\pi(.| s_t, y_t)$ does not allow writing the Bellman equation for the backward dynamics, and the expression of the backward reward is complex, as indicated in Lemma 3.1 of the revised manuscript.
>
> “_The computation of β_f∗ and β_b∗ needs to be done_”
>
> The analysis in Theorem C.1 provides the conditions that guarantee convergence. In practice, one could choose simple learning-rate settings with a confidence threshold such that these conditions are satisfied during the execution of the algorithm.
>
> “_Can’t equation (18) be expressed as a forward MDP?_”
>
> Thanks for the valuable comment. It is not possible to appropriately convert this controlled backward dynamics to a standard forward one. Specifically, if we flip the time-slot for the backward state we end up with a forward state which depends on the future actions that have not occurred yet and will happen in the far future. Hence, the states cannot be obtained by traversing onward over time.
>
> “_Enhancement of presentation … challenges in obtaining the analytical results?_”
>
> We have improved the presentation in the revised submission. Our work entails solving several challenges related to the analysis. First, we had to derive a backward version of the Bellman equations by identifying an action-coupled class of FB-MDPs. Second, we devised a shared entity for the actor agent which is updated based on the forward and backward stochastic rewards. To prove its convergence, we proposed a novel approach called episodic MCS-average. Finally, we analyzed convergence for merely smooth (and not convex) objective functions.

---

> > ### Comment · Reviewer_EeKJ · 2023-11-23
> >
> > Thank you for your response. It has resulted in a better understanding of the paper. Several concerns have been addressed in the revision.

---

### Official Review · Reviewer_g9Ad · 2023-10-31

**Soundness:** 3 good
**Presentation:** 2 fair
**Contribution:** 2 fair
**Rating:** 5
**Confidence:** 4

**Summary:**

This paper addresses sequential tasks that consider not only forward MDPs where states follow forward dynamics but also backward dynamics from which trajectories evolve in reverse-time order. In this setting, there can be conflicts between rewards from forward dynamics and those from backward ones. In this Forward-Backward Markov decision process (FB-MDP) setting, the authors propose an on-policy multi-objective reinforcement learning algorithm called Forward-Backward Multi-Objective Actor-Critic (FB-MOAC). The core idea is to use a multi-objective optimization technique so that the cumulative vector reward sum becomes a Pareto-optimal point. Numerical results show that the proposed FB-MOAC algorithm is applicable in the FB-MDP setting.

**Strengths:**

- This paper addresses sequential tasks on the Forward-Backward Markov decision process (FB-MDP) setting, which seems to provide a new perspective in (multi-objective) reinforcement learning.
- The paper provides some mathematical proofs regarding convergence to Pareto-optimal points.
- Based on previous works (Lemma 1), the proposed methodology gives technical soundness.

**Weaknesses:**

1. The paper is not easy to follow. There are many notational errors in the core formula. I recommend checking them to improve the overall readability.

2. The most critical weakness is that there are no baseline algorithms to compare with the proposed method in experiments. Readers cannot verify how much the FB-MOAC performs well.

3. Another issue is about the setting: FB-MDP. When can we consider this backward dynamics setting? It was hard to understand. Could the authors provide an example or detailed explanation regarding the necessity of considering backward dynamics?

4. Related to the above 3, the experiment part is hard to understand. Readers may not be familiar with wireless communication areas. Providing high-level pictures describing the environment is recommended.

5. Regarding reproducibility, it would be better to provide anonymous source code to verify the proposed algorithm.

**Questions:**

Please check the above weakness part. Additional questions are as follows.

6.  There is another issue about Lemma 1 which is the core foundation of FB-MOAC. Lemma 1 tells us about the Pareto-optimal convergence, but not about 'which Pareto-optimal point' to converge to. In most multi-objective RL works, there is a preference function (either linear or non-linear) that the designer cares about. Can Lemma 1 explain the characteristics of the converged Pareto-optimal point?

7. Conditions of Corollary 1 should be stated more precisely. There are inverse matrix operations and KKT conditions should be carefully considered.

8. Does saving past model parameters raise any memory issues of the algorithm?

---

> ### Author Response · Authors · 2023-11-17
>
> “_The paper is not easy to follow. There are many notational errors in the core formula_”
>
> Thanks for the valuable feedback. We carefully reviewed the notation and fixed some inconsistencies in the revised manuscript.
>
> “_There are no baseline algorithms to compare_”
>
> We agree that the random approach is not enough for comparison. Therefore, we have included two additional schemes for comparison purposes:Least Frequently Used (LFU), a rule-based approach widely used in the literature on caching; and F-MDP, a learning-based algorithm that replaces the backward MDP with the forward-only formulation.
>
> “_When can we consider this backward dynamics setting?_”
>
> The need for FB-MDPs depends on the problem environment under consideration. We have included an example in the context of computation offloading in the revised submission to motivate the applicability of FB-MDPs to a different scenario. Specifically, mobile devices offload computationally intensive tasks to an edge server, which processes them according to its computational capacity. The edge server has a buffer to store the tasks that cannot be immediately processed, which happens when there is no spare computational capacity. The new example has been mentioned in the introduction and detailed in the appendix.
>
> “_Providing high-level pictures describing the environment is recommended_”
>
> Thank you for your insightful comment. The hybrid experiment considered a heterogeneous network with two different node tiers: Base-stations (BSs) and helper-nodes (HNs). There exists a set of users that request files from a database containing $N$ different files. These files are requested based on a time-varying file popularity distribution. The network serves the users with a hybrid content-delivery scheme by employing both BSs and HNs. BSs are connected to the core-network and HNs are equipped with caches to proactively store files. The hybrid scheme is based on multicast transmissions from HNs and unicast transmissions from BSs. The HNs cache most popular files and cooperatively transmit the cached files across the network. Conversely, the BSs individually serve the users that request files. To do so, the BS first fetches the file from the core network and then sends it to the user. Transmissions are repeated until the data are successfully received by the user.
> Considering the popularity of content and the outage probability, a forward dynamics is found for the request probability which is expressed in Eq. (18). As an important metric for the transmission schemes, we consider the expected latency for successful content reception. Due to the outage possibility, a backward dynamics is extracted for the expected latency presented in Eq. (19). Then, based on the content popularity and expected latency, we are considering three conventional metrics to optimize the network performance. These metrics are as follows. A quality-of-service metric that indicates how much the users are satisfied, the bandwidth consumption that measures the required bandwidth of the whole network and the overall expected latency that shows the latency for all file requests.
>
> “_provide anonymous source code_”
>
> This is indeed an interesting point. We are reviewing and improving the clarity of the code used for the paper and are committed to anonymously publish it by the end of the discussion phase.
>
> “_Lemma 1 does not tell about which Pareto-optimal point it converges_”
>
> Thank you for the precise remark. Indeed, this Lemma does not specify which Pareto-optimal solution the devised algorithm converges to. However, the algorithm is guaranteed to converge to one of the Pareto-optimal points. As for our proposed method, we have proved convergence for both cases of strongly-convex and Lipschitz-smooth objectives and shown that all the forward and backward expected losses monotonically decrease as the iteration increases (Theorem C.1). A conventional approach for the multi-objective problems is scalarization, namely, by first constructing a scalar reward and then applying a single-objective algorithm. However, this approach makes the solution highly dependent on the selected scalarization technique. Instead, our approach aims to learn a scale-independent multi-objective learning approach. Furthermore, the considered methodology enables us to analyze the convergence of the proposed algorithm.
>
> “_Conditions of Corollary 1 should be stated more precisely._”
>
> We have modified this Corollary to more precisely express these conditions, according to the comment.
>
> “_Does saving past model parameters raise any memory issues of the algorithm?_”
>
> The FB-MOAC algorithm does not rely on storing the whole history of the model parameters: only the previous parameter is needed for the purpose of policy update. Our implementation ran on a laptop and did not experience any memory-related issues during numerical simulations.

---

> > ### Comment · Reviewer_g9Ad · 2023-11-21
> > **Thanks for the reply**
> >
> > Thank you for your response.
> >
> > I carefully reviewed the feedback and observed that several concerns have been addressed in the revision.
> >
> > However, a significant concern still remains regarding "Lemma 1 does not specify which Pareto-optimal point it converges to." In the field of multi-objective RL, there are two subareas: single-policy and multi-policy. When focusing on a specific scalarization function or utility, single-policy techniques are employed. If the goal is to obtain the entire approximation of the Pareto front (e.g., convex coverage set), then multi-policy techniques are adopted.
> >
> > It seems that the proposed method does not align with either of these settings. On the one hand, the algorithm is proven to converge to 'a' Pareto-optimal point but does not specify which Pareto-optimal point it converges to, i.e., which utility (either linear or non-linear) it optimizes. In some cases, understanding the optimized utility is crucial. On the other hand, the proposed algorithm does not consider covering the entire approximation of the Pareto front. In certain scenarios, covering various Pareto points (with linear preference) is essential. Moreover, it is unclear what the main factor is that differentiates the converged optimal point and how much of the (true) Pareto front it can cover.
> >
> > In summary, the proposed algorithm fails to (i) specify the characteristics of the converged Pareto point and (ii) guarantee coverage of (most of) the Pareto front. This aspect still raises concerns about the contribution of the paper.

---

> > > ### Author Response · Authors · 2023-11-22
> > >
> > > “_Lemma 1 does not specify which Pareto-optimal point it converges to._”
> > >
> > > Thanks for raising this point which allows us to clarify this aspect.
> > > In the context of multi-objective optimization problems, every Pareto-optimal solution necessarily adheres to the Karush-Kuhn-Tucker (KKT) conditions. However, not every solution satisfying the KKT conditions is a Pareto-optimal solution. Hence, obtaining a solution that merely satisfies the KKT conditions does not guarantee global optimality. To address this, we have incorporated Lemma 3.1 into our algorithm to systematically learn the combination of different objectives such that a Pareto-optimal solution can be obtained.
> > > Considering that the forward and backward rewards are solely coupled within the action space, we developed our algorithm based on a single-policy approach. While our algorithm shares similarities with conventional single-policy methods, it has the advantage of ensuring the convergence to globally (locally) Pareto-optimal solutions through a scale-insensitive manner – regardless of how the rewards are scaled, convergence is guaranteed.
> > > The characteristics of the obtained Pareto solutions are stated in Appendix C, and briefly summarized in Section 4.1. Specifically, for the strongly-convex objectives, a globally non-dominated Pareto solution is guaranteed, whereas for the only-smooth objectives, a Pareto-critical solution [1] (or a locally non-dominated Pareto solution) can be obtained. This implies that, for the strongly-convex objectives, no other solution exists that could improve one objective without worsening the others, and for the smoothness objectives, this holds locally. This distinguishes the characteristics of obtained Pareto solutions for the cases of strongly-convex objectives and smooth objectives.
> > > Given that our algorithm assures the convergence to a Pareto-optimal solution with a scale-insensitive manner, a key advantage is the ability to achieve the Pareto front by adjusting the actor initialization as well as by repeating the training procedure because of the stochastic policy. This is aligned with other multi-policy approaches [2] in which some hyper-parameters need to be altered to obtain the Pareto front.
> > > We will make some modifications in the revised paper to address this concern and provide a clearer explanation of this aspect.
> > >
> > > [1] Zhou, S., Zhang, W., Jiang, J., Zhong, W., Gu, J., & Zhu, W. (2022). On the Convergence of Stochastic Multi-Objective Gradient Manipulation and Beyond. In NeurIPS.
> > >
> > > [2] Abdolmaleki, A., Huang, S. H., et al. (2020). A Distributional View on Multi-Objective Policy Optimization. In Proceedings of the 37th International Conference on Machine Learning (pp. 11-22). PMLR.
> > >
> > >
> > > “_Code Reproducibility: it would be better to provide anonymous source code to verify the proposed algorithm_”.
> > >
> > > The first version of source code can be found by this link:   https://anonymous.4open.science/r/FB-MOAC-v1/README.md

---

> > > > ### Comment · Reviewer_g9Ad · 2023-11-23
> > > > **Thanks**
> > > >
> > > > Thanks for the additional comments. Although the comment attempted to address my concern about the contribution, I believe it is a repetition of the previous reply. The paper neither (i) specifies the non-decreasing utility regarding which the converged Pareto point optimizes nor (ii) guarantees coverage of (most of) the Pareto front "without repeating the training procedure". This aspect continues to raise concerns about the contribution of the paper.
> > > >
> > > > Nonetheless, with the two replies, most of the other concerns have been appropriately addressed. Therefore, I have adjusted my rating accordingly.

---

### Official Review · Reviewer_AwDv · 2023-11-04

**Soundness:** 3 good
**Presentation:** 3 good
**Contribution:** 2 fair
**Rating:** 5
**Confidence:** 3

**Summary:**

This paper introduces Forward-Backward Multi-Objective Reinforcement Learning (FB-MORL), an approach for addressing a special family of multi-task control problems. It presents the concept of the Forward-Backward Markov Decision Process (FB-MDP). Unlike the existing RL algorithms, this work primarily focuses on environments that cannot be modeled solely by a forward Markov decision process. This paper derives the policy gradient in the FB-MDP setting and proposes FB-MOAC, a stochastic actor-critic implementation of the policy gradient approach. In the policy update step, the author employs Monte Carlo Sampling (MCS) with an exponential moving average called episodic MCS-average to estimate the gradient. The proposed method is evaluated in wireless communication environments, and an ablation study highlights the importance of backward optimization and episodic MCS-average. The paper also provides theoretical analysis of FB-MOAC, demonstrating its convergence capability.

**Strengths:**

- The main contributions of this paper are mainly two-fold:
1. To the best of my knowledge, this paper offers the first formal study on the forward-backward MDPs and the resulting learning problem.
2. Accordingly, from the perspective of multi-objective RL, this paper proposes FB-MOAC, which is a policy-gradient-based actor critic method. This is built on the derived policy gradient in the FB-MDP setting.
- Convergence results of FB-MOAC are also provided (convergence to optimal total return under strict convexity and convergence to a local optimum in the general case).
- FB-MOAC is empirically evaluated on a hybrid delivery environment against a randomized baseline. A brief ablation study is also provided.

**Weaknesses:**

- The formulation of forward-backward MDP could be further justified. Despite that there is a whole line of research on backward SDEs, the use of the coupled forward and backward dynamics in the context of Markov processes and RL is quite rare and shall be justified. A motivating example early on in the paper would be helpful for readers to better appreciate this problem setting.

- The FB-MOAC algorithm is mainly built on Lemma 3.1, which suggests a first-order iterative descent approach to solve the general multi-objective optimization problems. However, this does not guarantee that an optimal policy could be achieved under FB-MOAC in general. While the authors did provide some convergence results in Section 5.1 and Appendix B, the best that FB-MOAC could achieve is convergence to a first-order stationary point (as the RL objective in general is not strictly convex).

- Based on the above, one important missing piece is the characterization of an optimal policy or the optimal value function in FB-MDPs. In the standard forward MDP, the widely known Bellman optimality equations offer a salient characterization of what an optimal value function and an optimal policy shall look like. It remains unclear what such conditions shall look like in the FB-MDP setting. In my opinion, further efforts on this is needed to design an algorithm that provably finds an optimal policy in general.

- Another related concern is on the policy class needed in the FB-MDP setting. In the standard forward (tabular) MDP setting, it is known that the class of (randomized) stationary policies is large enough to contain an optimal policy. However, it is rather unclear whether this is still true in FB-MDPs. However, this paper appears to directly presume that the policy is stationary. More discussion on this would be helpful and needed.

- Regarding the experiments of the hybrid delivery problem, while I could understand that this problem could be formulated as an FB-MDP (for keeping track of the latency), an alternative forward MDP formulation could still be used to solve this problem (for example, instead of keeping track of the latency of file n, the agent could look at the “time since the first transmission of file n”, which could be compatible with the standard forward MDP). Then, one natural question is: is there any benefit of using an FB-MDP instead of the (simpler) forward MDP formulation?

- Moreover, the experimental results are not very strong for two reasons: (1) It is unclear how far the performance of FB-MOAC is from an optimal policy (this is also related to one of my comments above). (2) The baseline (a simple randomized policy) is certainly not very strong. Given the plethora of research on unicast and multicast control in the wireless networking community, I would expect that there are some more competitive benchmark methods included in the experiments, either rule-based or learning-based. Also, it would be good to strengthen the experimental results by evaluating FB-MOAC on other RL environments. Otherwise, the application scope of FB-MOAC is somewhat limited.

**Questions:**

Please see the above for the main questions.

Here are some additional detailed questions:
- In Eq. (18), shall the 1/2 in the second term be just 1 (as Ln(t) denotes the latency)?

- While the whole Section 5.2.1 is used to describe the environment setup, it is still somewhat a bit difficult to parse (probably due to the use of terminology). For example,
    - How to define the event of multicast outage?
    - And accordingly how to derive the probability of this event?
    - How does the “request process” of each user work?

- The referencing numbers in the caption in Figure 1 and Figure 2 appear misplaced.

---------- Post-Rebuttal ----------

Thank the authors for the detailed response. Some of my concerns have been eased or addressed (e.g., added motivating examples for justifying FB-MDP and additional baselines).

---

> ### Author Response · Authors · 2023-11-17
>
> “_forward-backward MDP could be further justified_”
>
> We have included an example in the context of computation offloading in the revised submission to motivate the applicability of FB-MDPs to a different scenario. Specifically, mobile devices offload computationally intensive tasks to an edge server, which processes them according to its computational capacity. The edge server has a buffer to store the tasks that cannot be immediately processed, which happens when there is no spare computational capacity. The new example has been mentioned in the introduction and detailed in the appendix.
>
> “_Lemma 3.1 suggests a first-order iterative descent_”
>
> We appreciate the remark and the opportunity to clarify. The convergence of optimization algorithms based on this Lemma can be proven under the assumption of Lipschitz-smooth functions (Fliege et al., 2019)*. Our paper proves that for two cases: convexity and smoothness alone. Specifically, a locally Pareto-optimal solution with a convergence rate of $O(1/\sqrt{K})$ for the Lipschitz-smooth case is guaranteed, where $K$ is the number of updates. Section 4.1 provides a concise summary of these results, whereas Appendix C details the related derivation.
> * J. Fliege, A. I. F. Vaz & L. N. Vicente. “Complexity of gradient descent for multiobjective optimization,” Optimization Methods and Software, 34:5, 949-959, 2019.
>
> “_Bellman optimality equations offer a salient characterization_”
>
> The Bellman optimality equation characterizes the solution of dynamic programming algorithms and our work targets RL algorithms for action-coupled FB-MDPs. However, we here derive an alternative equation for FB-MDPs, namely, the Bellman Pareto-optimality equation. Please refer to the Lemma 3.1 of the revised manuscript and the following Remark.
>
> “_the class of stationary policies is large enough to contain an optimal policy_”
>
> We consider the forward and backward transition probabilities as stationary. We also assume that the forward and backward rewards are only coupled within the action space. Hence, the forward and backward value functions are stationary. Therefore, designing an optimal policy does not need to consider the time or the previous history of forward states. The same also holds for the backward reward. The evolution of the backward state is optimally determined by the forward state throughout the action space. As a result, a non-stationary policy distribution is not beneficial to jointly optimize the forward and backward rewards.
>
> “_forward MDP formulation could still be used to solve this problem_”
>
> Thanks for your suggestion. The backward reward is a part of the environment in the considered experiment. Therefore, replacing only that with another one would lead to a sub-optimal policy. In line with your suggestion, we consider an additional formulation to replace the backward dynamics. For this, we utilize this fact that minimizing both the outage probability and time-slot duration result in minimizing the overall latency, based on Eq.(19). Please refer to the revised manuscript for the additional results of the new formulation.
>
> “_ experimental results are not strong _”
>
> We point out that the experiments already considered two different cases: a hybrid and a multicast scheme. They differ from each other in terms of the system model, action parameters, reward function, and MDPs. Moreover, we do agree that the random approach is not a compelling baseline. Hence, we have included two additional schemes for comparison purposes: Least Frequently Used (LFU), a rule-based approach widely used in the literature on caching; and F-MDP, a learning-based algorithm that replaces the backward MDP with the forward-only formulation mentioned above.
>
> “_In Eq.(18), shall the 1/2 in the second term be just 1_”
>
> Thanks for the precise comment. The 1/2 factor is used under the assumption that requests in a time-slot are submitted according to a uniform distribution, corresponding to completely uncoordinated operations among users.
>
> “_How to define the event of multicast outage?_”
>
> The multicast outage happens when a user cannot successfully receive the requested file because the received signal quality is too low. The probability that such an outage occurs is given by $\mbox{Pr} ( B \log_2(1+\gamma)< R )$, where $\gamma$ is the Signal-to-Noise Ratio associated with the user, $B$ the bandwidth associated with the multicast scheme, and $R$ the desired transmission rate.
>
> “_how to derive the probability_”
>
> A closed-form expression of the probability has been obtained in (Amidzadeh et al., 2022). The derivation relies on leveraging stochastic geometry under the assumption that the base-stations are distributed according to a Poisson process.
>
> “_How does the “request process” of each user work?_”
>
> The users send requests to retrieve files towards the BSs by uplink transmissions. Each BS is responsible to deliver the file to the users associated with it and by using the conventional unicast scheme.

---

### Meta-Review · Area_Chair_5yMk · 2023-12-08

**Metareview:**

This research introduces Forward-Backward Multi-Objective Reinforcement Learning (FB-MORL) for addressing multi-task control problems involving both forward and backward dynamics. FB-MORL is evaluated in wireless caching scenarios, demonstrating effectiveness through empirical results. The proposed stochastic actor-critic implementation, FB-MOAC, undergoes theoretical analysis, showcasing convergence capabilities. The study emphasizes the importance of backward optimization and episodic MCS-average through an ablation study.

The significance of this work lies in its two-fold contribution: it formally studies the novel Forward-Backward Markov Decision Process (FB-MDP) setting, providing mathematical proofs for convergence to Pareto-optimal points. The proposed methodology is technically sound, drawing support from prior works, particularly Lemma 1. Furthermore, the paper emphasizes practical relevance by applying FB-MORL to a real-world wireless communication problem, showcasing its effectiveness in addressing complex, real-world challenges in the domain. Overall, the research contributes theoretically and practically to multi-objective reinforcement learning within the FB-MDP framework.

On the other hand, it has some issues that must be addressed. The paper could benefit from a clearer explanation of the Forward-Backward Markov Decision Process (FB-MDP) setting, potentially with a real-world example for better understanding. Numerous notational errors throughout the paper impede readability and require correction. The absence of baseline algorithms in experiments limits the evaluation of the proposed FB-MOAC method. The paper hints at potential computational challenges without providing a comprehensive discussion. In the wireless communication experiment section, complexity may hinder reader understanding, and visual aids or clearer descriptions would be beneficial. Lastly, providing anonymous source code for the proposed algorithm is recommended for transparency and reproducibility. Addressing these aspects would enhance the paper's overall quality and practical relevance.

The post-rebuttal discussion underscores ongoing concerns regarding the convergence behavior of FB-MOAC in achieving Pareto optimality or the entire Pareto set. The uncertainty about FB-MOAC reaching beyond a first-order stationary point remains a significant issue. Additionally, questions persist about the necessity of the forward-backward MDP formulation, especially in comparison to a simpler forward MDP formulation in scenarios like multicast.
These concerns lead the reviewer to lean towards maintaining their original rejection recommendation.
The authors are encouraged to address these issues while preparing a new version of their paper.

**Justification For Why Not Higher Score:**

This paper is borderline, but the reviewers agree that it is not ready for publication.

**Justification For Why Not Lower Score:**

N/A

---

### Decision · Program_Chairs · 2024-01-16

Reject